# A Joint Spectro-Temporal Relational Thinking Based Acoustic Modeling Framework

## Abstract

Relational thinking refers to the inherent ability of humans to form mental impressions about relations between sensory signals and prior knowledge, and subsequently incorporate them into their model of their world. This ability plays a key role in human understanding of speech, yet it has not been a prominent feature in any artificial speech recognition systems. Recently, there have been some attempts to correct this oversight, but these have been limited to coarse utterance-level models that operate exclusively in the time domain. In an attempt to narrow the gap between artificial systems and human abilities, this paper presents a novel spectro-temporal relational thinking based acoustic modeling framework. Specifically, it first generates numerous probabilistic graphs to model the relations among consecutive speech segments across both time and frequency domains. These graphs are then coupled and transformed into latent representations for downstream tasks, during which meaningful spectro-temporal patterns formed by the co-occurrence of certain node pairs can be uncovered. Models built upon this framework outperform state-of-the-art systems with a 7.82% improvement in phoneme recognition tasks. In-depth analyses further reveal that our proposed relational thinking modeling mainly improves the model's ability to recognize vowel phonemes.

## 1 Introduction

Deep learning techniques have brought in substantial advancements into automatic speech recognition (ASR), making it one of the most promising means of human-machine communication (Hinton et al., 2012). However, most deep neural network (DNN) based speech recognition systems (Vinyals et al., 2012; Abdel-Hamid et al., 2014; Chan et al., 2016; Passricha & Aggarwal, 2019; Wang et al., 2020; Baevski et al., 2020; Gulati et al., 2020) have drawn limited inspiration from the way speech is processed and recognized by human brain (Bohnstingl et al., 2022), instead treating the process as a black-box. As a consequence, the performances of these systems still lag behind that of the human brain (Malik et al., 2021). Recognizing the limitations inherent in current artificial systems, recent researches have endeavored to integrate biologically inspired mechanisms into existing DNN based systems, seeking to enhance interpretability and narrow the gap between artificial systems and the human brain (Dong & Xu, 2020; Bohnstingl et al., 2022).

Human minds are constantly and unconsciously filled with innumerable mental impressions pertaining to relations between current sensory signals and prior knowledge while hearing, seeing, smelling, etc. (Peirce, 2012). These mental impressions (i.e., *percepts*) are subsequently coupled and transformed into generalized understandings (i.e., *concepts*) (Mandler, 2007). This process, termed as *relational thinking*, is a fundamental human learning process that enables discerning meaningful patterns within the continuous flow of sensory data (Alexander, 2016). While humans rely on this inherent mechanism for speech comprehension (Birjandi & Sabah, 2012), artificial speech recognition systems have rarely employed it. The majority of the state-of-the-art systems, e.g., wav2vec2 (Baevski et al., 2020), have been developed using the transformer architectures (Vaswani et al., 2017), which basically employ attention mechanisms to capture dependencies between different parts of the sequence. However, these systems do not explicitly comprehend the relational information inherent in the sequence in the same way as the human brain. The attention mechanisms actually assess the significance of different parts of the sequential input when producing each entry of the output, allowing the model to focus on only pertinent information, as illustrated by Fig. 1 (a). In contrast, relational thinking captures the inherent relationships and interactions between various

pair-wise elements or features within the input sequence and estimates each entry of the output by aggregating all the pair-wise information, as shown in Fig. 1 (b). The information captured through relational thinking thus places a greater emphasis on the implications rooted in the co-occurrence of pairs of informative elements. This proves particularly beneficial to speech recognition, as certain pairs tend to appear jointly, for instance, the phonemes /m/ and /iy/ ("me", "autonomy", etc.).

However, such knowledge is not intrinsically captured by the attention mechanism prevalent in most current systems.

One of the few examples of the use of relational thinking models formulated this process in a conversational speech recognition scenario (Huang et al., 2020); the relational information acquired during the process was utilized as an additional input for the recognition task. However Huang et al. (2020) only investigated utterance-level relational

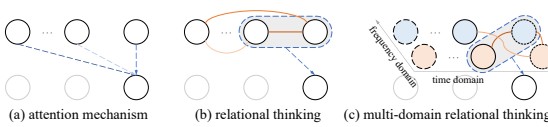

(a) attention mechanism    (b) relational thinking    (c) multi-domain relational thinking

Figure 1: Graphical illustrations of (a) attention mechanism, (b) conventional relational thinking, and (c) proposed joint spectro-temporal relational thinking. A darker line indicates greater importance (i.e., with a larger weight).

information in conversational scenarios. In a distinct yet highly relevant realm, natural language processing, Xue et al. (2021) proposed an approach to relation extraction which focused on extracting relations between words. Both Huang et al. (2020) and Xue et al. (2021) modeled the relations either at the utterance-level or the word-level. However, humans process speech and language at the more granular level of phonemes (Dusan & Rabiner, 2005; Wingfield et al., 2017). Furthermore, existing works have modeled the relations among elements of the input sequence separated in time only, whereas humans process speech by jointly considering multiple domains (e.g., time, frequency, semantics, etc.) rather than focusing exclusively on relationships in the time domain (Jurafsky & Martin, 2000).

In this paper, we propose a novel joint spectro-temporal relational thinking based acoustic modeling framework. The novelty lies in four aspects. 1) In contrast to previous approaches that focused solely on temporal patterns, the proposed framework captures relations across both time and frequency domains of the sensory input (as illustrated by Fig. 1 (c)) using a collection of probabilistic graphs, and then transform the relational information involved in the graphs into a form that can be used by downstream tasks. 2) Our approach tackles real-world scenarios where the input and output sequences differ in length. To facilitate the training of the proposed framework, we develop a tractable loss that optimizes the variational lower bound for the model log-likelihood. 3) Models built upon our proposed framework outperform the state-of-the-art baseline, demonstrating a performance gain of up to 7.82% in phoneme recognition tasks. Further analysis shows that the performance gain primarily originates from the model's enhanced ability to recognize vowels. This enhancement mirrors human proficiency in recognizing vowels more readily than consonants (Meyer et al., 2006). We also uncover the relevance of the captured relations to phoneme groups, where the patterns involved in the relations exhibit more similarities for phoneme classes within the same group. Additionally, the generalizability of the proposed framework is validated by employing other types of acoustic features (e.g., MFCCs), where relational thinking modeling consistently benefits downstream tasks. 4) We theoretically analyze the differences between self-attention mechanisms and relational thinking. These details are provided in Appendix D for those interested in further exploration.

## 2 MODELING RELATIONAL THINKING

Previous relational thinking approaches have employed graphs to model relationships between entries (or time steps) of a sequence, where each entry has been regarded as a node in the graphs. The goal of such approaches is to capture meaningful pair-wise patterns over time using these graphs (as illustrated by Fig.1 (b)), and then aggregate and transform the relational information involved in the graphs into a latent form that can be interpreted by subsequent layers of the model. Consider a sensory input $\mathbf{H} = [\mathbf{h}_1, \ldots, \mathbf{h}_T]$ corresponding to $T$ time steps. As illustrated by Fig. 2, the relational thinking process is carried out via the following three steps (Huang et al., 2020):

*1) Perception:* We first construct an infinite number of graphs $\{\mathcal{G}^{(k)}\}_{k=1}^{+\infty}$, where $\mathcal{G}^{(k)}(\mathcal{V}^{(k)}, \mathcal{E}^{(k)})$ is the $k$-th *percept* graph, with $\mathcal{V}^{(k)}$ and $\mathcal{E}^{(k)}$ denoting the node set and edge set, respectively. Each $\mathbf{h}_i \in \mathbb{R}^{D_h}, i = 1, \ldots, T$ corresponds to a node $v_i^{(k)}$ in every percept graph $\mathcal{G}^{(k)}$, while each element

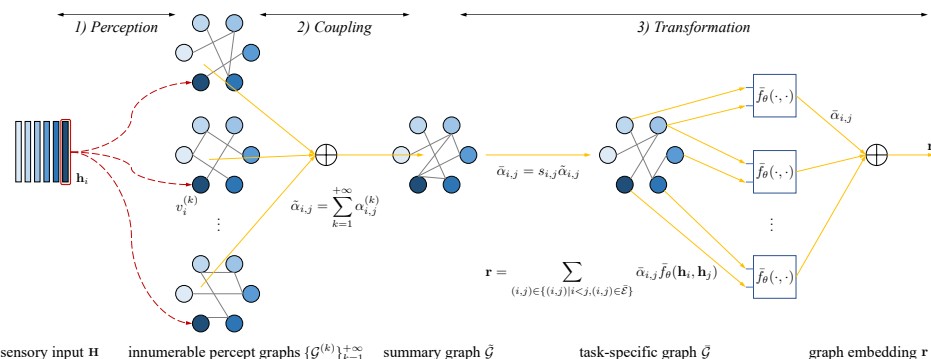

Figure 2: Modeling of relational thinking process.

$\alpha_{i,j}^{(k)}$ of the adjacency matrix $\mathbf{A}^{(k)}$ is associated with an edge $e_{i,j}^{(k)} \in \mathcal{E}^{(k)}$ between a pair of nodes $(v_i^{(k)}, v_j^{(k)})$ of $\mathcal{G}^{(k)}$. The value of $\alpha_{i,j}^{(k)}$ indicates the significance of the co-occurrence of node pair $(v_i^{(k)}, v_j^{(k)})$. Since the percepts form at an unconscious level of awareness (Rapp & Braasch, 2023), we assume that the probability of an edge's existence within the percept graphs is close to zero. To model this characteristic, we let the edge weights for the percept graphs follow a set of Bernoulli distributions, i.e., $\{\alpha_{i,j}^{(k)}\}_{k=1}^{+\infty} \sim \mathrm{Bern}(\lambda_{i,j})$, where the probability of edge existence $\lambda_{i,j} \to 0$.

*2) Coupling:* To aggregate the infinite number of percept graphs $\{\mathcal{G}^{(k)}\}_{k=1}^{+\infty}$, coupling is performed to derive an equivalent *summary* graph $\tilde{\mathcal{G}}$. In this graph, the original nodes $\mathbf{h}_1, \ldots, \mathbf{h}_T$ are preserved, while each edge $\tilde{\alpha}_{i,j}$ is obtained by summing up the corresponding edges over all percept graphs.

*3) Transformation:* Transformation converts the innumerable unconscious percepts into a recognizable notion of knowledge. Specifically, we first transform the summary graph $\tilde{\mathcal{G}}$, which represents the infinite number of percept graphs, into a *task-specific* graph $\bar{\mathcal{G}}$ by introducing transformation variables $s_{i,j}$ for each edge $\tilde{\alpha}_{i,j}$, such that $\bar{\alpha}_{i,j} = s_{i,j}\tilde{\alpha}_{i,j}$. Next, from $\bar{\mathcal{G}}$ we abstract a graph embedding $\mathbf{r}$ by summing up the embeddings of all node pairs weighted by $\bar{\alpha}_{i,j}$ as $\sum_{(i,j)\in\{(i,j)|i<j,(i,j)\in\bar{\mathcal{E}}\}} \bar{\alpha}_{i,j}\bar{f}_\theta(\mathbf{h}_i, \mathbf{h}_j)$. $\mathbf{r}$ is then ready for use in a specific downstream task.

The above described relational thinking modeling offers the unique ability to capture the co-occurrence of entries within the sensory input. The additional knowledge acquired during this process, which is not available from attention mechanisms (Vaswani et al., 2017), leads to further enhancement of the downstream task. More details about the modeling of relational thinking and the differences between relational thinking and attention mechanism are provided in Appendix C and Appendix D, respectively.

## 3 PROPOSED SPECTRO-TEMPORAL RELATIONAL THINKING FRAMEWORK

To exploit the range of information that are more readily accessible from different domains, (e.g., time domain, frequency domain, etc.), we propose a framework that models this process jointly across both the time and frequency domains (and more generally across the dimensions of any acoustic representation). This will enable a more comprehensive description of speech signals.

### 3.1 SPECTRO-TEMPORAL RELATIONAL THINKING BASED ACOUSTIC MODELING

The structure of the proposed acoustic modeling framework is depicted in Fig. 3. Given the raw waveform of a speech, we first employ the feature extraction module to calculate the acoustic feature vectors $\mathbf{c}_t \in \mathbb{R}^{D_c}, t = 1, \ldots, T$ corresponding to each of the time steps. Then, we re-organize them into a set of feature maps $\mathcal{C} = \{\mathbf{C}_1, \ldots, \mathbf{C}_T\}$ by forming each feature map with the current and the previous $w - 1$ time steps as $\mathbf{C}_t = [\mathbf{c}_{t-w+1}, \ldots, \mathbf{c}_t]^1$, guaranteeing the incorporation of causality.

---

[1]In a slight abuse of terminology, we refer to the feature space, in which $\mathbf{c}_1, \ldots, \mathbf{c}_T$ exist, as a *frequency* domain, although $\mathbf{c}_t$ can be an arbitrary type of acoustic feature.

$\mathbf{C}_t$ is subsequently used as the sensory input for relational thinking modeling of time step $t$. For time steps with $t < w$, specifically, $\mathbf{C}_t$ is padded with $\mathbf{0} \in \mathbb{R}^{D_c}$ such that all feature maps $\mathbf{C}_t, \forall t$ have the identical dimension of $D_c \times w$.

For the relational thinking module, every $\mathbf{C}_t$ is first smoothed and sub-sampled as

$$\check{\mathbf{C}}_t = \Xi(\mathbf{C}_t), \tag{1}$$

where $\Xi$ denotes a filtering operator. The function of $\Xi$ is to adjust the dimension of the original feature map $\mathbf{C}_t$, such that the resultant $\check{\mathbf{C}}_t$ has a dimension suitable for the subsequent spectro-temporal relational thinking modeling. Next, $\check{\mathbf{C}}_t$ is divided into a number of sub-feature maps as

$$\check{\mathbf{C}}_t = \begin{bmatrix} \boldsymbol{\Lambda}_{t,1,1} & \cdots & \boldsymbol{\Lambda}_{t,1,D^{(t)}} \\ \vdots & \ddots & \vdots \\ \boldsymbol{\Lambda}_{t,D^{(f)},1} & \cdots & \boldsymbol{\Lambda}_{t,D^{(f)},D^{(t)}} \end{bmatrix} \in \mathbb{R}^{D_c \times \tilde{w}}, \tag{2}$$

where every one of the total $u = D^{(f)} \times D^{(t)}$ sub feature maps $\boldsymbol{\Lambda}_{t,d^{(f)},d^{(t)}} \in \mathbb{R}^{D_s \times \tilde{w}_s}$ is ready to be mapped to a node within the percept graphs $\mathcal{G}_t^{(k)}$. As for the filtering $\Xi$ in (1), we explain its necessity with the example in Fig. 4. For the perception step of time domain modeling illustrated by Fig. 4 (a), each $\mathbf{c}_t$ from a time step can be directly mapped to a node in the percept graphs, with the number of nodes in a graph corresponding to the number of time steps ($w = 7$) included in $\mathbf{C}_t$. However, as per the spectro-temporal modeling, each node in the percept graphs encompasses information in both the time and frequency domains, spanning over $\tilde{w}_s$ and $D_s$, respectively. As illustrated by Fig. 4 (b), given $D_c = 6$, $w = 7$, and $u = 6$, it is not possible to evenly divide the $6 \times 7$ feature map $\mathbf{C}_t$ into

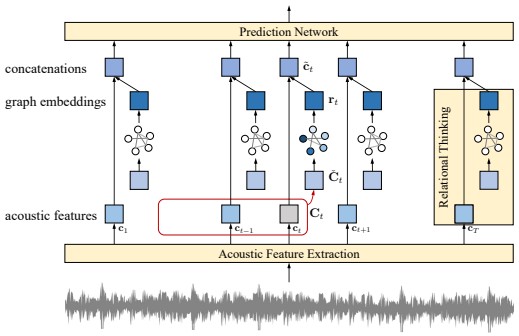

Figure 3: Spectro-temporal relational thinking based acoustic modeling framework.

2 rows and 3 columns, or 3 rows and 2 columns of sub-feature maps in the two red blocks in the figure. As a result, adjustments for the dimension of the original feature map $\mathbf{C}_t$ is necessary. We implement $\Xi$ with a temporal convolution in the proposed framework. Furthermore, we can define $\Delta_t := \check{w}/\tilde{w}_s, \Delta_f := D_c/D_s, \Delta_t, \Delta_f \in \mathbb{N}^+$ as the *resolutions* of relational thinking modeling in time and frequency domains, respectively. A higher resolution $\Delta_t$ or $\Delta_f$ indicates a more fine-grained capture of relations across the corresponding domain. By sequentially performing the perception, coupling, and transformation steps (as described by (7)–(12) in Appendix C) toward $\check{\mathbf{C}}_t$ for each time step $t$, we can obtain a sequence of graph embeddings $\mathbf{r}_1, \ldots, \mathbf{r}_T$.

By concatenating each $\mathbf{r}_t$ with the corresponding acoustic feature vector $\mathbf{c}_t$, we then obtain a more comprehensive speech representation

$$\tilde{\mathbf{c}}_t = [\mathbf{c}_t^T, \mathbf{r}_t^T]^T \tag{3}$$

for each time step. The sequence of the concatenated representations $\tilde{\mathbf{c}}_1, \ldots, \tilde{\mathbf{c}}_T$ is lastly fed into a prediction network (e.g., a linear projection) for the ultimate recognition task.

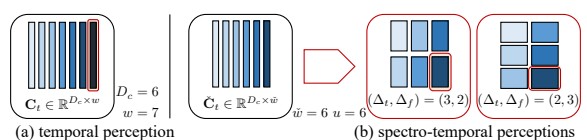

Figure 4: Perception step of (a) temporal and (b) spectro-temporal relational thinking modeling.

In the proposed spectro-temporal relational thinking modeling, a node pair refers to a spectro-temporal *pattern* formed by the co-occurrence of two sub feature maps within an *interval* (i.e., the temporal span covered by a sub-feature map $\boldsymbol{\Lambda}_{t,d^{(f)},d^{(t)}}$) or across intervals. Therefore, by incorporating both time and frequency domains, we are able to capture not only the relations between time intervals, but also the relations across different *frequency bands* within an interval or across intervals.

When modeling relational thinking for each time step $t$, it is essential to work with a feature map $\mathbf{C}_t$ that has a sufficiently wide context (i.e., with a sufficiently large $w$). This ensures that there

is enough local context available for effective relational thinking modeling. Therefore, in the proposed relational thinking framework, we take into account a context spanning at least 3 consecutive phonemes. This is in line with tri-phone models employed in HMM based acoustic models in the past (Jurafsky & Martin).

Also note that for a given number of nodes $u$ to be included in the graphs, there exist multiple choices for the resolutions $(\Delta_t, \Delta_f)$. As illustrated by the two solutions for the example in Fig. 4 (b), given $u = 6$, we can obtain either $(\Delta_t, \Delta_f) = (3, 2)$ or $(\Delta_t, \Delta_f) = (2, 3)$ for the spectro-temporal perception. Additionally, we can obtain two more solutions for the resolutions, i.e., $(6, 1)$ and $(1, 6)$, which in fact correspond to the temporal-only modeling and spectro-only modeling within a single domain. Variations in the resolution settings can have different effects on the performance of the downstream task. This aspect will be discussed in detail in Section 5.1.

## 3.2 TRAINING RELATIONAL THINKING BASED MODELS

For sequence modeling tasks like speech recognition, a common challenge arises from the varying lengths of the input and output sequences. This requires the loss function capable of managing such variations in sequence lengths. While Huang et al. (2020) and Chung et al. (2015) only considered the scenarios where the input and output sequences have equal lengths, our proposed spectro-temporal relational thinking framework is designed to handle more general scenarios where input and output sequences can have varying lengths. However, a tractable loss function is required to enable the training of our proposed framework. Given the complexity introduced by the random processes governing the generation of the graph edges (as detailed by (7)–(12) in Appendix C), direct optimization of the model log-likelihood $\log p(y|\mathcal{C})$ is infeasible. Instead, we employ the variational lower bound $\mathcal{L}$ (Sohn et al., 2015), by optimizing which log-likelihood can be also maximized:

$$\log p(y|\mathcal{C}) \geq \mathbb{E}_{q(\tilde{\mathcal{A}}, \mathcal{S}|\mathcal{C})}[\log p(y|\mathcal{C}, \tilde{\mathcal{A}}, \mathcal{S})] - \text{div}(q(\tilde{\mathcal{A}}, \mathcal{S}|\mathcal{C})\|p(\tilde{\mathcal{A}}, \mathcal{S}|\mathcal{C})) = \mathcal{L}, \tag{4}$$

where $\text{div}(\cdot\|\cdot)$ represents the KL divergence. In our proposed framework, we have two sets of variational latent variables that require optimization: $\tilde{\mathcal{A}} = \{\tilde{\mathbf{A}}_1, \ldots, \tilde{\mathbf{A}}_T\}$ and $\mathcal{S} = \{\mathbf{S}_1, \ldots, \mathbf{S}_T\}$, representing the adjacency matrices of the summary graphs and the graph transformation variable matrices for all time steps, respectively. $q(\tilde{\mathcal{A}}, \mathcal{S}|\mathcal{C})$ denotes the approximate posterior for $p(\tilde{\mathcal{A}}, \mathcal{S}|\mathcal{C}, y)$, while $p(\tilde{\mathcal{A}}, \mathcal{S}|\mathcal{C})$ represents the prior. For the case where input and output sequences have equal lengths (Huang et al., 2020; Chung et al., 2015), the prediction term in (4) can be decomposed into a frame-wise form as $\mathbb{E}_{q(\tilde{\mathcal{A}}, \mathcal{S}|\mathcal{C})}[\log p(y|\mathcal{C}, \tilde{\mathcal{A}}, \mathcal{S})] = \sum_{t=1}^{T} \mathbb{E}_{q(\tilde{\mathcal{A}}, \mathcal{S}|\mathcal{C})}[\log p(y_t|\mathbf{C}_t, \tilde{\mathcal{A}}, \mathcal{S})]$. However, it does not generalize to the case where input and output sequences are of different lengths. This forces us to recover $y$ using $\mathcal{C}, \tilde{\mathcal{A}}, \mathcal{S}$ throughout all time steps (see (17) in Appendix E). On the other hand, according to Nan et al. (2023), since $p(\tilde{\mathcal{A}}, \mathcal{S}|\mathcal{C}) = \prod_{t=1}^{T} p(\tilde{\mathbf{A}}_t, \mathbf{S}_t|\mathbf{C}_t)$, the KL divergence term in (4) can be decomposed as $\sum_{t=1}^{T} \text{div}(q(\tilde{\mathbf{A}}_t, \mathbf{S}_t|\mathbf{C}_t)\|p(\tilde{\mathbf{A}}_t, \mathbf{S}_t|\mathbf{C}_t))$, where $q(\tilde{\mathbf{A}}_t, \mathbf{S}_t|\mathbf{C}_t)$ and $p(\tilde{\mathbf{A}}_t, \mathbf{S}_t|\mathbf{C}_t)$ denote the approximate posterior and prior for time step $t$, respectively. Given that each element $s_{i,j}^{(t)}$ of $\mathbf{S}_t$ is conditioned on the Binomial variable $\tilde{\alpha}_{i,j}^{(t)}$ for the same edge of the $t$-th summary graph $\tilde{\mathcal{G}}_t$ (as indicated by (11) in Appendix C), we can further derive the KL divergence term for each time step $t$ as

$$\begin{aligned}
\text{div}(q(\tilde{\mathbf{A}}_t, \mathbf{S}_t|\mathbf{C}_t)\|p(\tilde{\mathbf{A}}_t, \mathbf{S}_t|\mathbf{C}_t)) = \sum_{(i,j)\in\tilde{\mathcal{E}}_t} \{&\text{div}(q(\tilde{\alpha}_{i,j}^{(t)}|\mathbf{C}_t)\|p(\tilde{\alpha}_{i,j}^{(t)}|\mathbf{C}_t)) \\
&+ \mathbb{E}_{q(\tilde{\alpha}_{i,j}^{(t)}|\mathbf{C}_t)}[\text{div}(q(s_{i,j}^{(t)}|\tilde{\alpha}_{i,j}^{(t)}, \mathbf{C}_t)\|p(s_{i,j}^{(t)}|\tilde{\alpha}_{i,j}^{(t)}, \mathbf{C}_t))]\}.
\end{aligned} \tag{5}$$

More details on the training of the proposed framework are provided in Appendix E, where we obtain a computationally tractable form of loss function that optimizes (4).

## 4 EXPERIMENTAL SETTINGS

**Goals** To gain insights into how the proposed model could aid downstream tasks, we aim to answer the following five questions:

**Q1:** *Does the proposed joint spectro-temporal modeling provide additional information that further benefits downstream tasks when compared to pure temporal or spectral modeling?*

**Q2:** *Is it more beneficial to model a larger context in the time domain or frequency domain?*

**Q3:** *Does the temporal span for relational thinking modeling affect the model's performance in downstream tasks?*

**Q4:** *Does relational thinking provide additional benefits beyond what the attention mechanism has achieved for downstream tasks?*

**Q5:** *Does the proposed framework consistently offer advantages across different types of acoustic features?*

**Dataset** We evaluate our proposed acoustic modeling framework in a general phoneme recognition downstream task. The TIMIT dataset (Garofolo et al., 1993) is used for training and evaluation, since it provides precise annotations for the start and end instants of each phoneme within an utterance, allowing for comprehensive analyses that lead to in-depth understanding of the proposed models. To recover the target phoneme sequence $y$, we use the best path decoding method (Graves et al., 2006). The phoneme error rate (PER) is employed for system evaluation.

**Settings** We employ the pre-trained wav2vec2 BASE (Baevski et al., 2020) for feature extraction and apply the proposed relational thinking modeling on top of it. Given that the majority (over 96%) of 3 consecutive phonemes in TIMIT has a duration shorter than 400 ms, we let $\mathbf{C}_t$ consist of 20 consecutive frames (i.e., $w = 20$, spanning a duration of 405 ms), such that the relations associated with 3 consecutive phonemes can be modeled. We include 8 nodes in each percept graph, i.e., $u = 8$. To answer Q1, we explore the time-only model w20-t8f1, the frequency-only model w20-t1f8, and the joint spectro-temporal models w20-t2f4 and w20-t4f2, by manipulating the resolutions for time and frequency domains as described in Section 3, with the naming of the models following the format w$w$-t$\Delta_t$f$\Delta_f$. To address Q3, we include another model w8-t2f4, with a different temporal span $w = 8$ for relational thinking modeling. For Q5, the effectiveness of the proposed model is further validated using MFCCs. Additional implementation details are available in Appendix F.

## 5 EXPERIMENTAL RESULTS AND ANALYSES

### 5.1 PHONEME RECOGNITION PERFORMANCE

**A1: Temporal vs. Spectral vs. Spectro-temporal** The performances of different models are compared in Table 1. We first fix the pre-trained parameters within the wav2vec2 module to eliminate the impact of variations in acoustic features. As shown in Table 1, the two joint spectro-temporal models, w20-t4f2 and w20-t2f4, outperform the temporal and spectral models, w20-t8f1 and w20-t1f8. This comparison clearly demonstrates the advantage of joint spectro-temporal modeling over the temporal or spectro modeling within a single domain. It is also evident that all the proposed relational thinking models (with 100.8M parameters in total) outperform the wav2vec2 baseline (with 94.4M parameters in total), with a relative reduction in PER ranging from 11.17% to 19.61%.

Table 1: Phoneme recognition performances of baselines and proposed models without fine-tuning.

| | | temporal | spectro | spectro-temporal | PER (%) dev | test |
|---|---|---|---|---|---|---|
| baseline | wav2vec2 | | | | 17.92 | 25.70 |
| proposed | w20-t8f1 | ✓ | | | 19.32 | 22.83 |
| | w20-t1f8 | | ✓ | | 16.14 | 21.76 |
| | w20-t4f2 | | | ✓ | 17.31 | **20.80** |
| | w20-t2f4 | | | ✓ | 14.02 | **20.66** |
| | w8-t2f4 | | | ✓ | 18.89 | 22.93 |

**A2: Trading off Temporal Context against Spectral Context** We compare the models with a higher resolution in frequency domain to those with a higher resolution in time domain. Specifically, two sets of models, {w20-t1f8, w20-t8f1} which model relations within a single (time or frequency) domain, and {w20-t2f4, w20-t4f2} which model relations in both time and frequency domains, are respectively compared. As illustrated by Table 1, in both sets, the model with higher frequency domain resolution (w20-t2f4 or w20-t1f8) exhibit superiority over its counterpart with higher time domain resolution (w20-t4f2 or w20-t8f1). This suggests that there might be potential benefits in modeling relations across frequency bands in greater detail by setting a higher frequency domain resolution compared to focusing more on time domain relations.

Figure 5: Relations learned by spectro-temporal relational thinking. Relational thinking evaluates the importance of the co-occurrence of a pair of nodes, representing a novel type of information beyond the assessment of individual nodes as typically done by attention mechanism. A pair of nodes is of more importance when the edge connecting them attains a larger value of $\bar{\alpha}_{i,j}^{(t)}$, as indicated by the arrows.

**A3: Impact of Temporal Span** To further understand the impact of the temporal span for relational thinking modeling, i.e., the value of $w$ for every $\mathbf{C}_t$, on the performance of downstream task, we compare two proposed models with relational thinking modeled throughout 20 and 8 consecutive time steps, respectively. In other words, relational thinking is performed throughout temporal spans corresponding to triphones and monophones in the two models, respectively.

We set the time and frequency resolutions to (2, 4) for both models. As shown in Table 1, the w8-t2f4 model, which incorporates relational information singly associated with the current phoneme at each time step, outperforms the wav2vec2 baseline with a 10.78% reduction in PER. However, when compared to the w20-t2f4 model, which incorporates relational information associated with the current and 2 preceding phonemes, it suffers a 10.99% drop in performance. This suggests that certain spectro-temporal patterns associated with consecutive phonemes contribute to further improving the prediction performance for the current phoneme.

Table 2: Phoneme recognition performances of baselines and proposed models in terms of PER (%) over TIMIT dataset.

|  |  | dev | test |
|---|---|---|---|
| baseline | CNN + TD-filterbanks (Zeghidour et al., 2018) | 15.6 | 18.0 |
|  | PASE+ (Ravanelli et al., 2020) | – | 17.2 |
|  | Li-GRU + fMLLR (Ravanelli et al., 2018) | – | 14.9 |
|  | wav2vec (Schneider et al., 2019) | 12.9 | 14.7 |
|  | vq-wav2vec (Baevski et al., 2019) | 9.6 | 11.6 |
|  | wav2vec2 (Baevski et al., 2020) | 7.26 | 9.98 |
| proposed | w20-t4f2 | 6.18 | **9.26** |
|  | w20-t2f4 | 6.23 | **9.20** |

**A4: Comparison with SOTA** The proposed models are compared with the transformer (more essentially, self-attention mechanism) based wav2vec2 baseline (Baevski et al., 2020) and other state-of-the-art systems (Zeghidour et al., 2018; Ravanelli et al., 2020; 2018; Schneider et al., 2019; Baevski et al., 2019) in Table 2. For the proposed models and the baseline, the (wav2vec2) feature extraction module is jointly optimized (i.e., fine-tuned) during training. Our proposed spectro-temporal models, w20-t4f2 and w20-t2f4, significantly outperform all the counterparts, specifically yielding 7.21% and 7.82% relative improvements of PER over the wav2vec2 baseline in the test dataset, respectively, revealing the additional advantages offered by relational thinking modeling compared to self-attention mechanism in enhancing speech representation.

**A5: Generalization to Other Acoustic Features** We also train a relational thinking based model using MFCCs (referred to as MFCC-RT-w20-t2f4) and compare it with an MFCC baseline implemented with a simple linear projection. Detailed configurations of the two models can be found in Appendix F.2. As shown in Table 3, the proposed MFCC-RT-w20-t2f4 model significantly outperforms the MFCC baseline, achieving a 14.36% reduction in PER over the test set. This validates that our proposed relational thinking modeling can generalize to sequential inputs composed of various types of acoustic features, providing additional relational information that consistently benefits downstream tasks.

Table 3: Phoneme recognition performances of baseline and proposed model using MFCCs.

|  |  | dev | test |
|---|---|---|---|
| baseline | MFCC | 39.80 | 47.90 |
| proposed | MFCC-RT-w20-t2f4 | 39.58 | **41.02** |

## 5.2 LEARNED RELATIONAL INFORMATION

The proposed relational thinking modeling enables us to infer relationships amongst different regions of the feature map without using any prior relational annotations during training. To illustrate the learned relational information, we randomly select a sample from the TIMIT test set, feed it into

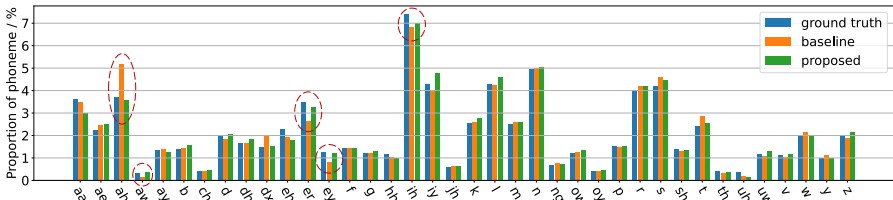

Figure 6: Proportions of recognized phoneme classes by baseline and proposed w20-t2f4 model. Ground truth reveals the actual proportions of all phoneme classes in the TIMIT test set. The proportions of vowel classes recognized by the proposed model align more closely with the ground truth proportions, suggesting the proposed model's better performance in recognizing vowels.

a proposed relational thinking model, and visualize the inferred task-specific graphs for 4 consecutive time steps out of the total $T$ time steps in Fig. 5. In each graph, a red curve represents an edge $\bar{\alpha}_{i,j}^{(t)}$ that connects a specific pair of sub feature maps from $\check{\mathbf{C}}_t$. The intensity of an edge's color corresponds to the regularized value of $\bar{\alpha}_{i,j}^{(t)}$ ranging from 0 to 1. For each time step, the respective task-specific graph clearly reveals the intricate relations among different sub feature maps of $\check{\mathbf{C}}_t$. As can be observed, all the task-specific graphs are relatively sparse, with only a few edges having large values of $\bar{\alpha}_{i,j}^{(t)}$ (as indicated in the figure). This observation aligns with our discussion in Section 3.1, indicating that certain spectro-temporal patterns (i.e., the co-occurrence of certain sub feature maps) are more important to the ultimate task than many others which are less meaningful.

We further explore the captured relational information by analyzing the edges $\bar{\alpha}_{i,j}^{(t)}$ of the learned task-specific graphs across different phoneme sub-groups (e.g., vowels, fricatives, nasals, etc.) in the frame-wise phoneme classification tasks. As shown in Fig. 12, the captured relations between different regions of the feature map, i.e., the edges $\bar{\alpha}_{i,j}^{(t)}$ of the task-specific graph, exhibit more similarities for phoneme classes within the same sub-group. However, the captured relations between phoneme classes from different sub-groups vary significantly. This suggests that the proposed model can discern and learn the intrinsic characteristics of various phoneme classes. More details can be found in Appendix G.2.

## 5.3 ANALYSIS OF PERFORMANCE OF DIFFERENT PHONEME GROUPS

We carry out additional analyses to gain a deeper understanding of how the proposed models enhance phoneme recognition performance. It is expected that the proposed relational thinking based models demonstrate superior advantages over the baseline in recognizing vowel phonemes. This is because vowel phonemes tend to have longer durations than non-vowel phonemes, allowing the relational thinking module to capture more significant relational information, which in turn benefits the downstream task. To this end, we separately investigate the performances of the models in recognizing vowels and non-vowels.

Intuitively, a good recognizer should produce a distribution of recognized phoneme classes that closely matches the ground truth distribution. To assess this, we calculate the proportion of each phoneme class among all the recognized phonemes in the test set for both the wav2vec2 baseline and the proposed w20-t2f4 model. These proportions are depicted in Fig. 6, where the ground truth proportions of phoneme classes in the test set are also provided. As can be observed, there are significant differences in the distributions of vowel classes (e.g., /ah/, /aw/, /er/, /ey/, /ih/) recognized by the two models, especially for the classes circled out in Fig. 6. In particular, the proportions of vowel classes recognized by the proposed model are more consistent with the ground truth proportions, with an average absolute difference of 0.23 pp. While the baseline shows a much higher average absolute difference of 0.35 pp (refer to Appendix G.1.1 for more details). This indicates that the proposed model exhibits superior performance in recognizing vowels.

Next, we conduct separate analyses of the errors made by both models in recognizing vowels and non-vowels. To be specific, given the recognition result of each model for a test sample, i.e., a sequence of recognized phonemes, we extract all the vowels/non-vowels from it and create a new sequence by combining the extracted phonemes with the original order preserved. This allows us

to formulate recognized vowel/non-vowel sequences. For example, we can obtain a vowel sequence [/ix/, /ah/, /ix/, /ae/, /ix/, /ix/, /ux/, /iy/, /ux/] from [/w/, /ix/, /dcl/, /s/, /ah/, /tcl/, /ch/, /ix/, /n/, /ae/, /kcl/, /t/, /ix/, /v/, /r/, /ix/, /f/, /y/, /ux/, /zh/, /el/, /bcl/, /b/, /iy/, /y/, /ux/, /s/, /f/, /el/]. The ground truth vowel/non-vowel sequences can be derived from the reference target sequence in the same way. To approximate the errors made by each model in recognizing vowels/non-vowels, we calculate the edit distance (Navarro, 2001) between the recognized vowel/non-vowel sequence and the corresponding ground truth counterpart for all test samples. Fig. 7 illustrates the distributions of edit distances between the recognized sequences and the ground truth sequences for all test samples.

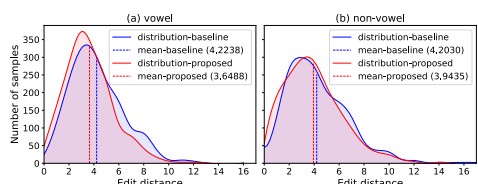

In Fig. 7 (a), which pertains to the performances of the two models in recognizing vowels, it is evident that the proposed model outperforms the baseline. The distribution of edit distances for the proposed model is significantly skewed towards the left, compared to that for the baseline, with the average edit distance for the proposed model (3.6488) much smaller than that for the baseline (4.2238). While for the performances of the two models in recognizing non-vowels as depicted in Fig. 7 (b), the proposed model only shows a slight improvement over the baseline. In this case, the distributions of errors made by the two models are closer compared to the case of recognizing vowels, where the average edit distances for the two models are 3.9435 and 4.2030, respectively.

Figure 7: Distributions of edit distances between recognized sequences and ground truth sequences.

When the biologically inspired relational thinking process is incorporated, the model's performance in recognizing vowels shows a more noticeable improvement compared to its performance in recognizing non-vowels. This finding also coincides with the results of a speech intelligibility test conducted with human listeners, as reported in Meyer et al. (2006). The test results suggested that vowel identification is a relatively easier task for humans compared to consonant identification. Additional analyses related to the phoneme recognition tasks can be referred to Appendix G.1.

## 5.4 SPEECH RECOGNITION WITH PROPOSED FRAMEWORK

The proposed spetro-temporal relational thinking modeling is further validated in speech recognition tasks and evaluated using word error rate (WER), aiming to demonstrate the generalizability of our proposed framework to other prevalent tasks. A word-level relational thinking model, built upon the proposed framework (as detailed in Appendix G.3), displays a 2.55% reduction in WER against the wav2vec2 baseline (Baevski et al., 2020) when language model is not applied. The incorporation of a 4-gram language model increases this reduction in WER to 3.23% (refer to Table 6 for details). These improvements imply that comprehending and utilizing the spectro-temporal relations associated with words also advantages the downstream speech recognition tasks, in that certain words tend to coherently and frequently appear together, such as "I am".

## 6 CONCLUSION

We propose a novel spectro-temporal relational thinking based acoustic modeling framework, where its core module is inspired by a fundamental human learning process. This framework is capable of capturing a unique form of pair-wise information, distinct from the assessment of individual nodes as performed by attention mechanism. Models constructed using this framework show state-of-the-art performance in phoneme recognition tasks. Further analysis conveys connections between the captured relations and phoneme groups, where the patterns involved in the relations exhibit more similarities for phoneme classes within the same group, while showing significant variations between phoneme classes from different groups. Our analysis also reveals that relational thinking modeling primarily enhances the model's ability to recognize vowels. Additionally, we demonstrate the generalizability of the proposed framework by applying other types of acoustic features and employing it for different downstream tasks, where relational thinking modeling consistently benefits downstream tasks. This study aims to pave a new pathway for integrating biologically inspired human learning processes into deep learning approaches, improving the model's capability in speech recognition and potentially its interpretability.

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

# A    FUTURE WORK

Given the success of the proposed framework in both phoneme-level and word-level recognition tasks, respectively, as well as the evidence suggesting that speech recognition in human brain is a holistic process that involves various types of linguistic units (e.g., phonemes, syllables, words, etc.) (Dusan & Rabiner, 2005), we can potentially extend the proposed single-level modeling to a multi-level (e.g., phoneme-level and word-level) modeling. Relying on such an approach, we would be able to simultaneously capture the relations associated with linguistic units across different levels, allowing us to explore their combined impact on the ultimate task.

# B    RELATED WORK

*1) Automatic Speech Recognition:* There have been extensive studies into automatic speech recognition (ASR) over the past decades. In a classic ASR pipeline, the acoustic model was constructed using a combination of Gaussian mixture models (GMMs) and a Hidden Markov model (HMM), where the GMMs were employed to model the observation probabilities associated with the hidden states, while the HMM was responsible for modeling the transition probabilities among the hidden state sequence (Mohamed et al., 2009; He et al., 2008). However, due to its limited representational capacity, the GMM-HMM model struggles to effectively model the streams of interacting knowledge sources in speech. Furthermore, the decoding and learning processes for the GMM-HMM model are feasible only when relying on the Markov assumption or conditional independence assumption (Mohamed et al., 2009). Driven by the advancements in computer hardware and deep learning techniques, ASR has then undergone a profound change. Following the initial adoption of deep neural networks (DNNs), which were introduced to replace GMMs for hidden state modeling (Mohamed et al., 2009; Dahl et al., 2011; Hinton et al., 2012), researchers then explored a range of neural network structures to enhance the performance of ASR systems. These included the use of one or a mixture of the convolutional neural networks (CNNs), recurrent neural networks (RNNs), transformers, etc. (Abdel-Hamid et al., 2014; Vinyals et al., 2012; Passricha & Aggarwal, 2020; Wang et al., 2020). Training and deployment processes for ASR systems were both largely simplified with the advent of end-to-end models. The connectionist temporal classification (CTC) approach enabled automatic detection of the alignment between the input and output sequences, eliminating the requirement for pre-segmented training data (Graves et al., 2006). The attention based encoder-decoder (AED) architecture further removed any assumptions about the dependency of the output sequence (Chorowski et al., 2015; Chan et al., 2016; Hou et al., 2017). To address the challenge of limited annotated data, self-supervised learning techniques have been incorporated. Models were initially pre-trained to learn general speech representations from large volumes of unannotated data, and were subsequently fine-tuned using limited annotated data for specific downstream tasks (Schneider et al., 2019; Baevski et al., 2019; 2020). Notably, the latest wav2vec2 framework has prominently improved the state-of-the-art performance in speech recognition tasks (Baevski et al., 2020). However, most existing systems have drawn limited inspiration from the way speech is processed and recognized by human brain (Bohnstingl et al., 2022), rendering their performance still lagging behind that of human brain (Malik et al., 2021). Our biologically inspired approach, in contrast, explicitly models the intrinsic relational thinking process that is critically associated with human speech processing. This in turn leads to further performance improvement.

*2) Relational Thinking Modeling:* Studies on the modeling of relational thinking process are still quite primitive. Huang et al. (2020) developed utterance-level relational thinking modeling for conversational speech recognition. Relying on a Bayesian deep learning method (Wang & Yeung, 2020), they modeled the perception process by generating an infinite number of probabilistic graphs representing the unconscious percepts, with each node within a graph referring to an utterance in a conversation. To address the computational challenges arising from the innumerable number of percepts, they obtained an analytical solution by creating an equivalent new graph that summarized the original ones. The relational information involved in this graph was further transformed into a form that could be used in downstream tasks (refer to Fig. 8 and Fig. 2 (a)). In the field of natural language processing, Xue et al. (2021) introduced a simplified modeling approach for relation extraction tasks. They extended a BERT-based language model by incorporating a latent multi-view graph created using a Gaussian graph generator. This graph aimed to capture various potential word-level relations between tokens. This approach differs from Huang et al. (2020) in terms of the task's

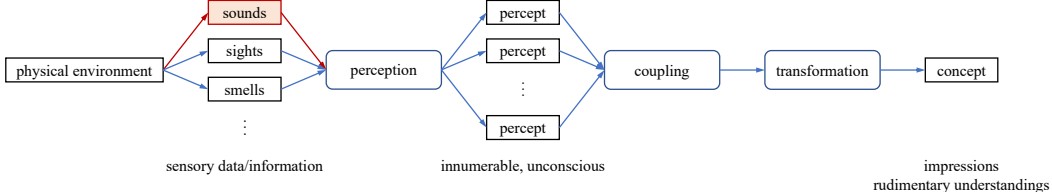

Figure 8: Illustration of relational thinking process in human brain (Rapp & Braasch, 2023).

objective. Specifically, in Xue et al. (2021), the goal was to explicitly learn the relations between tokens, where the relational information had been provided by the dataset and was adopted as labels. In contrast, Huang et al. (2020) focused on inferring relations among utterances without relying on any prior relational data. Besides, both Huang et al. (2020) and Xue et al. (2021) have modeled the relations among the input sequence solely throughout time domain. Nevertheless, human brain processes speech by jointly considering multiple domains (e.g., time, frequency, semantics, etc.) rather than focusing exclusively on time domain (Jurafsky & Martin, 2000). Our work presents a spectro-temporal relational thinking modeling, which can offer a more comprehensive representation of speech and further narrows the gap between artificial systems and human brain. Concerning the sensory signals from multiple sources, Xing et al. (2019) introduced a perception coordination network (PCN) designed to replicate the process of concept acquisition from multi-modal sensory signals (e.g., vision, audition, and touch) in human brain. Within the hierarchical structure of the network, fundamental features (e.g., colors, shapes, syllables, basic flavors) were initially transformed into a set of unimodal concepts. Subsequently, a multi-sensory integration step was implemented to associate these primary unimodal concepts. However, despite its potential, the PCN, formulated as a type of self-organizing network (Jantvik et al., 2011), still faces challenges when it comes to its practical application in specific downstream tasks.

## C  DETAILS OF RELATIONAL THINKING MODELING

The entire relational thinking process as described in Section 2 is demonstrated by Fig. 2. Consider a sensory input

$$\mathbf{H} = [\mathbf{h}_1, \ldots, \mathbf{h}_T] \tag{6}$$

corresponding to $T$ time steps. The relational thinking process can be primitively modeled throughout time domain with the following steps (Huang et al., 2020).

*1) Perception:* We first construct an infinite number of graphs $\{\mathcal{G}^{(k)}\}_{k=1}^{+\infty}$, where $\mathcal{G}^{(k)}(\mathcal{V}^{(k)}, \mathcal{E}^{(k)})$ is the $k$-th *percept* graph, with $\mathcal{V}^{(k)}$ and $\mathcal{E}^{(k)}$ denoting the node set and edge set, respectively. Each $\mathbf{h}_i, i = 1, \ldots, T$ corresponds to a node $v_i^{(k)}$ in every percept graph $\mathcal{G}^{(k)}$, while each element $\alpha_{i,j}^{(k)}$ of the adjacency matrix $\mathbf{A}^{(k)}$ is associated with an edge $e_{i,j}^{(k)} \in \mathcal{E}^{(k)}$ between node $i$ and node $j$ of $\mathcal{G}^{(k)}$.

Since the percepts attain an unconsciousness characteristic (Alexander, 2016), we assume that the probability of edge existence within the percept graphs is close to zero. Specifically, considering an infinite number of edges $\{\alpha_{i,j}^{(k)}\}_{k=1}^{+\infty}$ between node $i$ and node $j$, we sample as

$$\left\{\alpha_{i,j}^{(k)}\right\}_{k=1}^{+\infty} \sim \text{Bern}(\lambda_{i,j}), \tag{7}$$

where $\text{Bern}(\lambda_{i,j})$ is a Bernoulli distribution with probability of edge existence $\lambda_{i,j} \to 0$.

*2) Coupling:* Coupling aims to obtain a *summary* graph $\tilde{\mathcal{G}}$ that is capable of representing the infinite number of percept graphs $\{\mathcal{G}^{(k)}\}_{k=1}^{+\infty}$. In this graph, the original nodes $\mathbf{h}_1, \ldots, \mathbf{h}_T$ are kept. While since it is intractable to simply take a summation over all adjacency matrices $\{\mathbf{A}^{(k)}\}_{k=1}^{+\infty}$, we equivalently generate each edge of $\tilde{\mathcal{G}}$ upon sampling from a Binomial distribution, i.e.,

$$\tilde{\alpha}_{i,j} = \sum_{k=1}^{+\infty} \alpha_{i,j}^{(k)} \quad \Leftrightarrow \quad \tilde{\alpha}_{i,j} \sim \mathcal{B}(n, \lambda_{i,j}), \tag{8}$$

where $n \to +\infty$ and $\lambda_{i,j} \to 0$.

**Theorem 1.** *Let $\mathcal{B}(n, \lambda)$ denote a Binomial distribution with $n \to +\infty, \lambda \to 0$, and let $m = n\lambda$. There exists a Gaussian distribution $\mathcal{N}(m, m(1-m))$ that approximates $\mathcal{B}(n, \lambda)$ with a bounded approximation error, where*

$$m = \frac{1}{2} \left\{ 1 + \frac{2\sigma^2}{1 - 2\mu} - \left[ 1 + \left( \frac{2\sigma^2}{1 - 2\mu} \right)^2 \right]^{\frac{1}{2}} \right\} \tag{9}$$

*is derived from a Gaussian distribution $\mathcal{N}(\mu, \sigma^2)$ with $\mu < 1/2$.*

According to Theorem 1 (Huang et al., 2020), by letting $m_{i,j} = n\lambda_{i,j}$, we can bypass the direct parameterization of both the infinite $n$ and the near-zero $\lambda_{i,j}$, and find a tractable Gaussian proxy $\mathcal{N}(m_{i,j}, m_{i,j}(1 - m_{i,j}))$ for the Binomial distribution $\mathcal{B}(n, \lambda_{i,j})$ in (8).

*3) Transformation:* Transformation converts the innumerable unconscious percepts into a recognizable notion of knowledge. This indicates that we abstract an informative representation from the summary graph $\tilde{\mathcal{G}}$ for downstream tasks. Specifically, this transformation is designed as first weighting each edge $\tilde{\alpha}_{i,j}$ of $\tilde{\mathcal{G}}$ with a Gaussian variable $s_{i,j}$:

$$\bar{\mathbf{A}} = \mathbf{S} \odot \tilde{\mathbf{A}}, \tag{10}$$

where $\odot$ denotes the Hadamard product, $\mathbf{S}$ is the graph transformation matrix, $\tilde{\mathbf{A}}$ and $\bar{\mathbf{A}}$ are the adjacency matrices of $\tilde{\mathcal{G}}$ and the transformed graph $\bar{\mathcal{G}}$, respectively. $s_{i,j}$ is assumed to be conditioned on the corresponding edge $\tilde{\alpha}_{i,j}$ of $\tilde{\mathcal{G}}$, i.e.,

$$s_{i,j}|\tilde{\alpha}_{i,j} \sim \mathcal{N}(\tilde{\alpha}_{i,j}\mu_{i,j}, \tilde{\alpha}_{i,j}\sigma_{i,j}^2). \tag{11}$$

The transformed graph $\bar{\mathcal{G}}$ is named as a *task-specific* graph. Finally, a graph embedding $\mathbf{r}$ is extracted from $\bar{\mathcal{G}}$ as

$$\mathbf{r} = \sum_{(i,j) \in \{(i,j)|i<j,(i,j)\in\bar{\mathcal{E}}\}} \bar{\alpha}_{i,j} \bar{f}_\theta(\mathbf{h}_i, \mathbf{h}_j), \tag{12}$$

where $\bar{\mathcal{E}}$ is the edge set of $\bar{\mathcal{G}}$, and $\bar{f}_\theta(\cdot, \cdot)$ denotes a node pair embedding function (Kipf et al., 2018). Thus, the graph embedding $\mathbf{r}$ abstracted from $\bar{\mathcal{G}}$ is ready to be used as additional information for a specific downstream task.

# D  COMPARISON BETWEEN SELF-ATTENTION MECHANISM AND RELATIONAL THINKING

Self-attention is a technique designed to emulate cognitive attention processes. It has been widely adopted as a pivotal component of the transformer networks (Vaswani et al., 2017). In this mechanism, the weights and the representation of the relations involved are calculated as

$$\begin{cases} \alpha_{i,j} = \text{softmax}(\text{score}(\mathbf{W}_q\mathbf{h}_i, \mathbf{W}_k\mathbf{h}_j)), \quad i, j = t - w + 1, \ldots, t, \quad \text{score}(\mathbf{q}, \mathbf{k}) = \dfrac{\mathbf{k}^T\mathbf{q}}{\sqrt{|\mathbf{k}|}}, \\[2mm] \mathbf{e}_i = \displaystyle\sum_{j=t-w+1}^{t} \alpha_{i,j} f_v(\mathbf{h}_j), \quad i = t - w + 1, \ldots, t, \quad f_v(\mathbf{h}) = \mathbf{W}_v\mathbf{h}, \\[2mm] \mathbf{r}_t = \mathbf{e}_t. \end{cases} \tag{13}$$

For the ease of comparison, we reframe the relational thinking process (7)–(12) into the following form

$$\begin{cases} \bar{\alpha}_{i,j}^{(t)}, \quad i, j = t - w + 1, \ldots, t, \quad \bar{\alpha}_{i,j}^{(t)} \text{ is obtained by a generative process (7)–(11),} \\[2mm] \mathbf{e}_i^{(t)} = \displaystyle\sum_{j=t-w+1,j>i}^{t} \bar{\alpha}_{i,j} \bar{f}_\theta(\mathbf{h}_i, \mathbf{h}_j), \quad i = t - w + 1, \ldots, t, \quad \bar{f}_\theta(\mathbf{h}_i, \mathbf{h}_j) = \text{MLP}\left(\left[\mathbf{h}_i^T, \mathbf{h}_j^T\right]^T\right), \\[2mm] \mathbf{r}_t = \displaystyle\sum_{i=t-w+1}^{t} \mathbf{e}_i^{(t)}. \end{cases} \tag{14}$$

As indicated by (13) and (14), both the self-attention mechanism and relational thinking share a similar calculation structure. With either of these techniques, we eventually derive a representation $\mathbf{r}_t$ that captures the relations among different nodes (frames). This representation has a form of a weighted sum of embeddings.

However, the methods for determining the weights $\alpha_{i,j}$ in (13) and $\bar{\alpha}_{i,j}^{(t)}$ in (14) differ between the two techniques. In self-attention mechanism, a node $\mathbf{h}_i$ is initially projected into both a query space and a key space by $\mathbf{W}_q$ and $\mathbf{W}_k$, respectively. Then there is always a scoring process, where the score typically quantifies the relevance or importance of the key vector $\mathbf{W}_k \mathbf{h}_j$ concerning the query vector $\mathbf{W}_q \mathbf{h}_i$. While in relational thinking, the weights $\bar{\alpha}_{i,j}^{(t)}$ are generated through a generative process (7)–(11) as outlined in Appendix C. Each weight corresponds to an edge connecting two nodes in the task-specific graph.

Next, when comparing the second equations in (13) and (14), we can further observe notable differences in the embedding functions used by the two techniques. In self-attention mechanism, a single node is typically embedded using a linear transformation $f_v(\mathbf{h}) = \mathbf{W}_v \mathbf{h}$. In contrast, in relational thinking, a pair of nodes is embedded together using a network $\bar{f}_\theta(\cdot, \cdot)$. This distinction leads to a fundamental difference in the outcomes. Specifically, self-attention mechanism ultimately calculates a weighted sum of node embeddings, while relational thinking computes a weighted sum of node pair embeddings. Consequently, by incorporating relational thinking modeling, models gain the ability to effectively assess the importance of a pair of nodes to the downstream task, in addition to the capability of measuring the importance of individual nodes, which is already provided by the self-attention mechanism.

### D.1 RELATIONAL THINKING VS. STACKED SELF-ATTENTION MECHANISM

Consider a simplified 2-layer self-attention network w.l.o.g., where each layer $l$ comprises two nodes $\mathbf{h}_1^{(l)}$ and $\mathbf{h}_2^{(l)}$. According to (13), the calculation of nodes in the subsequent layer $l+1$ is as follows:

$$\left[\mathbf{h}_1^{(l+1)}, \mathbf{h}_2^{(l+1)}\right] = \mathbf{W}_v^{(l)} \left[\mathbf{h}_1^{(l)}, \mathbf{h}_2^{(l)}\right] \begin{bmatrix} \alpha_{1,1}^{(l)} & \alpha_{2,1}^{(l)} \\ \alpha_{1,2}^{(l)} & \alpha_{2,2}^{(l)} \end{bmatrix}. \tag{15}$$

As a result, the state of a node $\mathbf{h}_2^{(3)}$ after undergoing two layers of self-attention calculations is

$$\mathbf{h}_2^{(3)}$$
$$= \alpha_{2,1}^{(2)} \mathbf{W}_v^{(2)} \left(\alpha_{1,1}^{(1)} \mathbf{W}_v^{(1)} \mathbf{h}_1^{(1)} + \alpha_{1,2}^{(1)} \mathbf{W}_v^{(1)} \mathbf{h}_2^{(1)}\right) + \alpha_{2,2}^{(2)} \mathbf{W}_v^{(2)} \left(\alpha_{2,1}^{(1)} \mathbf{W}_v^{(1)} \mathbf{h}_1^{(1)} + \alpha_{2,2}^{(1)} \mathbf{W}_v^{(1)} \mathbf{h}_2^{(1)}\right) \tag{16a}$$

$$= \left(\alpha_{1,1}^{(1)} \alpha_{2,1}^{(2)} + \alpha_{2,1}^{(1)} \alpha_{2,2}^{(2)}\right) \mathbf{W}_v^{(2)} \mathbf{W}_v^{(1)} \mathbf{h}_1^{(1)} + \left(\alpha_{1,2}^{(1)} \alpha_{2,1}^{(2)} + \alpha_{2,2}^{(1)} \alpha_{2,2}^{(2)}\right) \mathbf{W}_v^{(2)} \mathbf{W}_v^{(1)} \mathbf{h}_2^{(1)}. \tag{16b}$$

Even though (16a) may exhibit a similar form to (12), particularly when we view $\mathbf{W}_v^{(2)} \left(\alpha_{1,1}^{(1)} \mathbf{W}_v^{(1)} \mathbf{h}_1^{(1)} + \alpha_{1,2}^{(1)} \mathbf{W}_v^{(1)} \mathbf{h}_2^{(1)}\right)$ and $\mathbf{W}_v^{(2)} \left(\alpha_{2,1}^{(1)} \mathbf{W}_v^{(1)} \mathbf{h}_1^{(1)} + \alpha_{2,2}^{(1)} \mathbf{W}_v^{(1)} \mathbf{h}_2^{(1)}\right)$ as node pair embedding functions from a linear family $\mathcal{F}\left(\mathbf{h}_1^{(1)}, \mathbf{h}_2^{(1)}\right)$, it is crucial to note that $\mathbf{h}_2^{(3)}$ is fundamentally still a weighted sum of node embeddings (as revealed by (16b)) rather than a weighted sum of node pair embeddings as obtained by relational thinking (12), where $\bar{f}_\theta(\mathbf{h}_i, \mathbf{h}_j)$ is an arbitrary node pair embedding function. Therefore, the stacked self-attention mechanism and relational thinking are not entirely equivalent, and models cannot solely rely on the self-attention mechanism to effectively assess the importance of a pair of nodes.

## E TRAINING OF PROPOSED FRAMEWORK

As revealed by (7)–(12), the proposed spectro-temporal relational thinking modeling involves a series of random processes that govern the generation of the graph edges. This characteristic classifies it as a variational model, rendering the infeasibility of directly optimizing the model log-likelihood $\log p(y|\mathcal{C})$, where $y$ denotes the target sequence. Instead, the variational lower bound $\mathcal{L}$ is typically employed (Sohn et al., 2015), by optimizing which log-likelihood can be maximized:

$$\log p(y|\mathcal{C}) \geq \mathbb{E}_{q(\tilde{\mathcal{A}}, \mathcal{S}|\mathcal{C})} \left[\log p\left(y \middle| \mathcal{C}, \tilde{\mathcal{A}}, \mathcal{S}\right)\right] - \operatorname{div}\left(q\left(\tilde{\mathcal{A}}, \mathcal{S} \middle| \mathcal{C}\right) \middle\| p\left(\tilde{\mathcal{A}}, \mathcal{S} \middle| \mathcal{C}\right)\right) = \mathcal{L},$$

where $\mathrm{div}(\cdot\|\cdot)$ represents the KL divergence. In our proposed framework, we have two sets of variational latent variables. $\tilde{\mathcal{A}} = \{\tilde{\mathbf{A}}_1, \ldots, \tilde{\mathbf{A}}_T\}$ and $\mathcal{S} = \{\mathbf{S}_1, \ldots, \mathbf{S}_T\}$ collect the adjacency matrices of the summary graphs and the graph transformation matrices for all time steps, respectively. $q(\tilde{\mathcal{A}}, \mathcal{S}|\mathcal{C})$ denotes the approximate posterior for $p(\tilde{\mathcal{A}}, \mathcal{S}|\mathcal{C}, y)$, while $p(\tilde{\mathcal{A}}, \mathcal{S}|\mathcal{C})$ represents the prior. The variational CTC loss (Nan et al., 2023) can be employed to optimize $\mathcal{L}$:

$$\tilde{\mathcal{L}} = \sum_{B \in F^{-1}(y)} \prod_{t=1}^{T} p\left(b_t \,\middle|\, \mathcal{C}, \tilde{\mathcal{A}}, \mathcal{S}\right) - \sum_{t=1}^{T} \mathrm{div}\left(q\left(\tilde{\mathbf{A}}_t, \mathbf{S}_t \,\middle|\, \mathbf{C}_t\right) \,\middle\|\, p\left(\tilde{\mathbf{A}}_t, \mathbf{S}_t \,\middle|\, \mathbf{C}_t\right)\right), \quad (17)$$

where $B = [b_1, \ldots, b_T]$ denotes an alignment between $\mathcal{C}$ and $y$ (Graves et al., 2006), $b_t \in \mathcal{W} \cup \{-\}$, $\mathcal{W}$ is the target vocabulary, and $F$ maps the paths $B$ with the same length as $\mathcal{C}$ to the target sequence $y$ by first merging the consecutive duplicated labels into one and then discarding the blanks "$-$". In (17), since $p(\tilde{\mathcal{A}}, \mathcal{S}|\mathcal{C}) = \prod_{t=1}^{T} p(\tilde{\mathbf{A}}_t, \mathbf{S}_t|\mathbf{C}_t)$, the KL divergence term in (4) is decomposed into a frame-wise form, where $q(\tilde{\mathbf{A}}_t, \mathbf{S}_t|\mathbf{C}_t)$ and $p(\tilde{\mathbf{A}}_t, \mathbf{S}_t|\mathbf{C}_t)$ denote the approximate posterior and prior for time step $t$, respectively. As can be observed, the loss function $\tilde{\mathcal{L}}$ consists of a prediction objective and a regularization objective. The prediction objective guides the model to recover the target sequence $y$, while the regularization objective encourages the model to keep its posterior distributions close to the corresponding priors for all time steps. Note that in contrast with Huang et al. (2020) and Chung et al. (2015), the prediction objective in (17) cannot be decomposed into a frame-wise form. To be specific, in these two works, the input sequence and the target sequence have the same length, i.e., they are aligned. This allows the prediction objective to be easily decomposed as $\sum_{t=1}^{T} \mathbb{E}_{q(\mathbf{Z}_t|\mathbf{X}_t)}[\log p(y_t|\mathbf{X}_t, \mathbf{Z}_t)]$. However, we formulated our problem (4) as a more general one, where the input and target sequences might not be aligned. This forces us to integrally recover the target sequence $y$ using the input feature maps $\mathcal{C}$ and the latent variables $\tilde{\mathcal{A}}, \mathcal{S}$ throughout all time steps, as described by (17).

As each element $s_{i,j}^{(t)}$ of $\mathbf{S}_t$ is conditioned on the Binomial variable $\tilde{\alpha}_{i,j}^{(t)}$ for the same edge of the $t$-th summary graph $\tilde{\mathcal{G}}_t$ (as indicated by (11)), the KL divergence terms in (17) can be further derived as

$$\begin{aligned}
&\mathrm{div}\left(q\left(\tilde{\mathbf{A}}_t, \mathbf{S}_t \,\middle|\, \mathbf{C}_t\right) \,\middle\|\, p\left(\tilde{\mathbf{A}}_t, \mathbf{S}_t \,\middle|\, \mathbf{C}_t\right)\right) \\
&= \sum_{(i,j) \in \tilde{\mathcal{E}}_t} \mathrm{div}\left(q\left(\tilde{\alpha}_{i,j}^{(t)}, s_{i,j}^{(t)} \,\middle|\, \mathbf{C}_t\right) \,\middle\|\, p\left(\tilde{\alpha}_{i,j}^{(t)}, s_{i,j}^{(t)} \,\middle|\, \mathbf{C}_t\right)\right) \\
&= \sum_{(i,j) \in \tilde{\mathcal{E}}_t} \mathrm{div}\left(q\left(\tilde{\alpha}_{i,j}^{(t)} \,\middle|\, \mathbf{C}_t\right) q\left(s_{i,j}^{(t)} \,\middle|\, \tilde{\alpha}_{i,j}^{(t)}, \mathbf{C}_t\right) \,\middle\|\, p\left(\tilde{\alpha}_{i,j}^{(t)} \,\middle|\, \mathbf{C}_t\right) p\left(s_{i,j}^{(t)} \,\middle|\, \tilde{\alpha}_{i,j}^{(t)}, \mathbf{C}_t\right)\right) \\
&= \sum_{(i,j) \in \tilde{\mathcal{E}}_t} \int_{\tilde{\alpha}_{i,j}^{(t)}} \int_{s_{i,j}^{(t)}} q\left(\tilde{\alpha}_{i,j}^{(t)} \,\middle|\, \mathbf{C}_t\right) q\left(s_{i,j}^{(t)} \,\middle|\, \tilde{\alpha}_{i,j}^{(t)}, \mathbf{C}_t\right) \log \frac{q\left(\tilde{\alpha}_{i,j}^{(t)} \,\middle|\, \mathbf{C}_t\right) q\left(s_{i,j}^{(t)} \,\middle|\, \tilde{\alpha}_{i,j}^{(t)}, \mathbf{C}_t\right)}{p\left(\tilde{\alpha}_{i,j}^{(t)} \,\middle|\, \mathbf{C}_t\right) p\left(s_{i,j}^{(t)} \,\middle|\, \tilde{\alpha}_{i,j}^{(t)}, \mathbf{C}_t\right)} d\tilde{\alpha}_{i,j}^{(t)} ds_{i,j}^{(t)} \\
&= \sum_{(i,j) \in \tilde{\mathcal{E}}_t} \left\{ \int_{\tilde{\alpha}_{i,j}^{(t)}} \int_{s_{i,j}^{(t)}} q\left(s_{i,j}^{(t)} \,\middle|\, \tilde{\alpha}_{i,j}^{(t)}, \mathbf{C}_t\right) ds_{i,j}^{(t)} q\left(\tilde{\alpha}_{i,j}^{(t)} \,\middle|\, \mathbf{C}_t\right) \log \frac{q\left(\tilde{\alpha}_{i,j}^{(t)} \,\middle|\, \mathbf{C}_t\right)}{p\left(\tilde{\alpha}_{i,j}^{(t)} \,\middle|\, \mathbf{C}_t\right)} d\tilde{\alpha}_{i,j}^{(t)} \right. \\
&\qquad\qquad \left. + \int_{\tilde{\alpha}_{i,j}^{(t)}} q\left(\tilde{\alpha}_{i,j}^{(t)} \,\middle|\, \mathbf{C}_t\right) \int_{s_{i,j}^{(t)}} q\left(s_{i,j}^{(t)} \,\middle|\, \tilde{\alpha}_{i,j}^{(t)}, \mathbf{C}_t\right) \log \frac{q\left(s_{i,j}^{(t)} \,\middle|\, \tilde{\alpha}_{i,j}^{(t)}, \mathbf{C}_t\right)}{p\left(s_{i,j}^{(t)} \,\middle|\, \tilde{\alpha}_{i,j}^{(t)}, \mathbf{C}_t\right)} ds_{i,j}^{(t)} d\tilde{\alpha}_{i,j}^{(t)} \right\} \\
&= \sum_{(i,j) \in \tilde{\mathcal{E}}_t} \left\{ \mathrm{div}\left(q\left(\tilde{\alpha}_{i,j}^{(t)} \,\middle|\, \mathbf{C}_t\right) \,\middle\|\, p\left(\tilde{\alpha}_{i,j}^{(t)} \,\middle|\, \mathbf{C}_t\right)\right) \right. \\
&\qquad\qquad \left. + \mathbb{E}_{q\left(\tilde{\alpha}_{i,j}^{(t)}|\mathbf{C}_t\right)} \left[ \mathrm{div}\left(q\left(s_{i,j}^{(t)} \,\middle|\, \tilde{\alpha}_{i,j}^{(t)}, \mathbf{C}_t\right) \,\middle\|\, p\left(s_{i,j}^{(t)} \,\middle|\, \tilde{\alpha}_{i,j}^{(t)}, \mathbf{C}_t\right)\right) \right] \right\}, \quad (18)
\end{aligned}$$

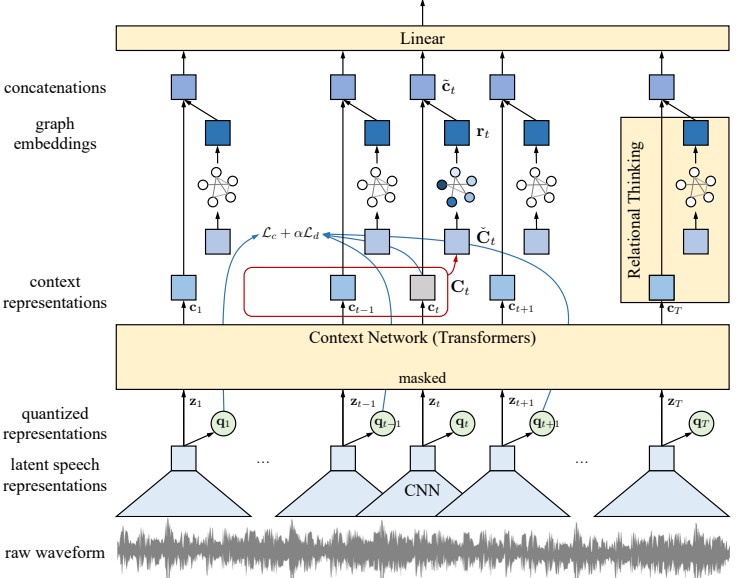

Figure 9: A phoneme recognition model implemented using the proposed spectro-temporal relational thinking based acoustic modeling framework. wav2vec2 is employed as the acoustic feature extraction module. The prediction network is implemented with a linear projection.

where

$$
\begin{aligned}
q\left(\left.\tilde{\alpha}_{i,j}^{(t)}\right| \mathbf{C}_t\right) &= \mathcal{B}\left(n^{(t)}, \tilde{\lambda}_{i,j}^{(t)}\right), \\
p\left(\left.\tilde{\alpha}_{i,j}^{(t)}\right| \mathbf{C}_t\right) &= \mathcal{B}\left(n^{(t)}, \tilde{\lambda}_{i,j}^{(t,0)}\right), \\
q\left(\left.s_{i,j}^{(t)}\right| \tilde{\alpha}_{i,j}^{(t)}, \mathbf{C}_t\right) &= \mathcal{N}\left(\tilde{\alpha}_{i,j}^{(t)}\mu_{i,j}^{(t)}, \tilde{\alpha}_{i,j}^{(t)}\sigma_{i,j}^{(t)2}\right), \\
p\left(\left.s_{i,j}^{(t)}\right| \tilde{\alpha}_{i,j}^{(t)}, \mathbf{C}_t\right) &= \mathcal{N}\left(\tilde{\alpha}_{i,j}^{(t)}\mu_{i,j}^{(t,0)}, \tilde{\alpha}_{i,j}^{(t)}\sigma_{i,j}^{(t,0)2}\right).
\end{aligned}
\tag{19}
$$

The KL divergences between two Gaussian distributions are readily simplified to the following closed-form:

$$
\operatorname{div}\left(q\left(\left.s_{i,j}^{(t)}\right| \tilde{\alpha}_{i,j}^{(t)}, \mathbf{C}_t\right)\middle\| p\left(\left.s_{i,j}^{(t)}\right| \tilde{\alpha}_{i,j}^{(t)}, \mathbf{C}_t\right)\right) = \frac{1}{2}\log\frac{\sigma_{i,j}^{(t,0)2}}{\sigma_{i,j}^{(t)2}} + \frac{\sigma_{i,j}^{(t)2} + \left(\mu_{i,j}^{(t)} - \mu_{i,j}^{(t,0)}\right)^2}{2\sigma_{i,j}^{(t,0)2}} - \frac{1}{2}, \tag{20}
$$

with $\tilde{\mathcal{E}}_t$ denoting the edge set of the $t$-th summary graph. Then, with the intractability of $n^{(t)} \to +\infty$, $\tilde{\lambda}_{i,j}^{(t)} \to 0$, and $\tilde{\lambda}_{i,j}^{(t,0)} \to 0$ eliminated by introducing two variables $m_{i,j}^{(t)} = n^{(t)}\tilde{\lambda}_{i,j}^{(t)}$ and $m_{i,j}^{(t,0)} = n^{(t)}\tilde{\lambda}_{i,j}^{(t,0)}$, respectively, we can follow the Theorem 2 in Huang et al. (2020) to further derive closed-form of KL divergences between two Binomial distributions in (5) as

$$
\operatorname{div}\left(q\left(\left.\tilde{\alpha}_{i,j}^{(t)}\right| \mathbf{C}_t\right)\middle\| p\left(\left.\tilde{\alpha}_{i,j}^{(t)}\right| \mathbf{C}_t\right)\right) < m_{i,j}^{(t)}\log\frac{m_{i,j}^{(t)}}{m_{i,j}^{(t,0)}} + \left(1 - m_{i,j}^{(t)}\right)\log\frac{1 - m_{i,j}^{(t)} + \frac{m_{i,j}^{(t)2}}{2}}{1 - m_{i,j}^{(t,0)} + \frac{m_{i,j}^{(t,0)2}}{2}}. \tag{21}
$$

Finally, by substituting (5), (18)–(21) into (17), we obtain a closed-form loss function, allowing direct optimization of the variational lower bound $\mathcal{L}$ of the model log-likelihood.

# F  DETAILED EXPERIMENTAL CONFIGURATIONS

## F.1  PROPOSED MODELS EMPLOYING WAV2VEC2 FOR ACOUSTIC FEATURE EXTRACTION

We apply our proposed acoustic modeling framework (as illustrated by Fig. 3) to a general phoneme recognition downstream task. The structure of the phoneme recognition models implemented using the proposed framework is shown in Fig. 9.

### F.1.1  ACOUSTIC FEATURE EXTRACTION MODULE

Since wav2vec2 is one of the state-of-the-art frameworks for extracting speech representations (Baevski et al., 2020), it is employed as the acoustic feature extraction module in our proposed models. Here, we give necessary descriptions about the structure of wav2vec2, while more details about its implementation and training can be found in Baevski et al. (2020).

The wav2vec2 framework mainly consists of three modules, namely, a feature encoder, a context network, and a quantization module. As illustrated by Fig. 9, the feature encoder $f : \mathcal{X} \rightarrow \mathcal{Z}$ contains a series of temporal convolutional blocks. The input of this module is the raw audio waveform $\mathcal{X}$ corresponding to an utterance, while its output is a sequence of latent speech representations $\mathbf{z}_1, \ldots, \mathbf{z}_T$ for the $T$ time steps. The output of the feature encoder $\mathcal{Z}$ is then fed into the subsequent context network $g : \mathcal{Z} \rightarrow \mathcal{C}$, which is built up with the serialization of several transformers (Vaswani et al., 2017). This module generates the contextualized representations $\mathbf{c}_1, \ldots, \mathbf{c}_T$, which capture information from the entire speech representation sequence. Relying on a product quantization process (Jegou et al., 2010; Baevski et al., 2019), the quantization module $q : \mathcal{Z} \rightarrow \mathcal{Q}$ discretizes the outputs of the feature encoder into $\mathbf{q}_1, \ldots, \mathbf{q}_T$, which are from a finite set of speech representations. To pre-train the wav2vec2 framework, the following loss function is optimized:

$$\mathcal{L}_{\text{pt}} = \mathcal{L}_c + \alpha \mathcal{L}_d, \tag{22}$$

where

$$\mathcal{L}_c = -\log \frac{\exp\left(\frac{\text{sim}(\mathbf{c}_t, \mathbf{q}_t)}{\kappa}\right)}{\sum_{\tilde{\mathbf{q}} \in \mathbf{Q}_t} \exp\left(\frac{\text{sim}(\mathbf{c}_t, \tilde{\mathbf{q}})}{\kappa}\right)} \tag{23}$$

and

$$\mathcal{L}_d = \frac{1}{GV} \sum_{g=1}^{G} -H\left(\bar{p}_g\right) = \frac{1}{GV} \sum_{g=1}^{G} \sum_{v=1}^{V} \bar{p}_{g,v} \log \bar{p}_{g,v} \tag{24}$$

are the contrastive loss and the diversity loss, respectively. The contrastive loss requires the model to identify the true quantized latent speech representation for a masked time step $t$ within a set of distractors, while the diversity loss is designed for the sake of increasing the use of the quantized codebook representations (Baevski et al., 2020).

In our proposed models, we directly employ the pre-trained wav2vec2 BASE[2] as the acoustic feature extraction module.

### F.1.2  SPECTRO-TEMPORAL RELATIONAL THINKING MODULE

For the relational thinking module, we first investigate the duration distribution of 3 consecutive phonemes within the TIMIT dataset (Garofolo et al., 1993). As shown in Fig. 10, the majority (over 96%) of these 3-phoneme sequences have a duration shorter than 400 ms. On the technical side, the wav2vec2 framework defines a frame width of 25 ms and a frame stride of 20 ms. Aiming to create a feature map that is capable of effectively modeling relations among 3 consecutive phonemes for the majority of cases, we should let it consist of at least $w = 20$ frames, covering a time span of 405 ms. As a result, the feature map is designed as $\mathbf{C}_t = [\mathbf{c}_{t-19}, \ldots, \mathbf{c}_t] \in \mathbb{R}^{768 \times 20}$, where 768 corresponds to the dimension of the context representations generated by wav2vec2 BASE. The kernel width and kernel stride for the temporal convolution in (1) are set to 5 and 2, respectively, leading to $\check{\mathbf{C}}_t \in \mathbb{R}^{768 \times 8}$ and the number of nodes included in the percept graphs being $u = 8$. Therefore, we can derive four different sets of resolution settings $(\Delta_t, \Delta_f)$ for time and frequency domains, i.e., (8, 1), (4, 2), (2, 4), and (1, 8), respectively.

---

[2]https://huggingface.co/facebook/wav2vec2-base.

Figure 10: Duration distribution of 3 consecutive phonemes within TIMIT dataset. Over 96% of the 3-phoneme sequences have a duration shorter than 400 ms. 20 wav2vec2 frames cover a time span of 405 ms.

For the approximate posterior $q(\tilde{\alpha}_{i,j}^{(t)}|\mathbf{C}_t)$, we sample $\tilde{\alpha}_{i,j}^{(t)}$ from the Gaussian proxy $\mathcal{N}(m_{i,j}^{(t)}, m_{i,j}^{(t)}(1 - m_{i,j}^{(t)}))$ of $\mathcal{B}(n^{(t)}, \tilde{\lambda}_{i,j}^{(t)})$ by letting $m_{i,j}^{(t)} = n^{(t)}\tilde{\lambda}_{i,j}^{(t)}$. However, according to Theorem 1, before calculating the parameter $m_{i,j}^{(t)}$ of the Gaussian proxy with (9), we have to first learn from the input $\mathbf{C}_t$ another Gaussian distribution $\mathcal{N}(\tilde{\mu}_{i,j}^{(t)}, \tilde{\sigma}_{i,j}^{(t)2})$ with $\tilde{\mu}_{i,j}^{(t)} < 1/2$. Here, we employ two multi-layer perceptrons (MLPs) for the inference of $\tilde{\mu}_{i,j}^{(t)}$ and $\tilde{\sigma}_{i,j}^{(t)}$, respectively, taking $\mathbf{C}_t$ as their inputs. Each MLP has a hidden layer with 128 nodes. For the corresponding prior $p(\tilde{\alpha}_{i,j}^{(t)}|\mathbf{C}_t)$, we learn the parameter $m_{i,j}^{(t,0)}$ with an MLP (with 128 nodes in the hidden layer), taking as input the feature map $\mathbf{C}_t$. Note that we cannot directly draw samples $\tilde{\alpha}_{i,j}^{(t)}$ from the Gaussian proxy $\mathcal{N}(m_{i,j}^{(t)}, m_{i,j}^{(t)}(1 - m_{i,j}^{(t)}))$ here. Instead, we have to involve the re-parameterization trick (Kingma & Welling, 2013). Specifically, we first draw an auxiliary variable $\gamma$ from $\mathcal{N}(0, 1)$. Then, we obtain $\tilde{\alpha}_{i,j}^{(t)}$ as $\tilde{\alpha}_{i,j}^{(t)} = m_{i,j}^{(t)} + (m_{i,j}^{(t)}(1 - m_{i,j}^{(t)}))^{1/2}\gamma$, enabling the parameter $m_{i,j}^{(t)}$ to be differentiable.

For the approximate posterior $q(s_{i,j}^{(t)}|\tilde{\alpha}_{i,j}^{(t)}, \mathbf{C}_t)$, i.e., $\mathcal{N}(\tilde{\alpha}_{i,j}^{(t)}\mu_{i,j}^{(t)}, \tilde{\alpha}_{i,j}^{(t)}\sigma_{i,j}^{(t)2})$, we adopt two MLPs (with 128 nodes in the hidden layer) to predict $\mu_{i,j}^{(t)}$ and $\sigma_{i,j}^{(t)}$, respectively, taking $\mathbf{C}_t$ as inputs. The parameters of the corresponding prior, $\mu_{i,j}^{(t,0)}$ and $\sigma_{i,j}^{(t,0)}$, can be obtained similarly. Again, we rely on the re-parameterization trick to sample $s_{i,j}^{(t)}$.

The node pair embedding function $\bar{f}_\theta(\cdot, \cdot)$ in (12) is implemented with an MLP, where the hidden layer has 128 nodes. Context representations with respect to the two nodes are concatenated and then fed into the MLP. The output dimension of $\bar{f}_\theta(\cdot, \cdot)$ is 32. As a result, we eventually obtain a graph embedding $\mathbf{r}_t \in \mathbb{R}^{32}$ for each time step, together with the concatenated representation $\tilde{\mathbf{c}}_t \in \mathbb{R}^{800}$.

Following the protocol outlined in Lee & Hon (1989), we keep all the original 62 possible phoneme classes during training, but collapse them to 39 classes during evaluation. The phoneme error rate (PER) is employed as the metric for recognition performance:

$$\text{PER} = \frac{1}{N}\sum_{i=1}^{N}\frac{\text{dist}\,(\hat{y}_i, y_i)}{|y_i|}, \tag{25}$$

where dist $(\hat{y}_i, y_i)$ represents the edit distance (also known as Levenshtein distance) between two phoneme sequences $\hat{y}_i$ and $y$ (Navarro, 2001), $\hat{y}_i$ is the prediction for $y_i$, $|y_i|$ denotes the cardinality of $y_i$, and $N$ is the number of samples.

### F.2    MODELS USING MFCCS AS INPUTS

To demonstrate the generalizability of our proposed framework, we also train a set of models using the MFCCs as acoustic features. The MFCC baseline is implemented with a simple linear projection, taking as input the MFCC feature vectors from all time steps, instead of the context representations generated by wav2vec2 BASE. In contrast, the relational thinking based MFCC model, referred to as MFCC-RT-20-t2f4, further computes the graph embeddings for all time steps using the feature maps $\mathbf{C}_t$ obtained from the MFCC feature vectors, following the procedures outlined in Section 3.1. Here, the width of $\mathbf{C}_t$ is set to $w = 20$, and the resolutions for time and frequency domains are (2, 4). The MFCC feature vectors and the graph embeddings are concatenated before fed into the linear projection. Since MFCC calculations are deterministic, fine-tuning is not required for these two models.

## G    ADDITIONAL EXPERIMENTAL RESULTS AND ANALYSES

### G.1    ANALYSES FOR PHONEME RECOGNITION TASKS

#### G.1.1    PROPORTIONS OF PHONEME CLASSES

Table 4 presents the proportions of phoneme classes recognized by the wav2vec2 baseline and the proposed w20-t2f4 model, along with the ground truth proportions of phoneme classes in the test set. Values within parentheses indicate the absolute differences between the proportions recognized by the two models and the ground truth proportions. The average absolute difference between the proportions of recognized vowel classes by the baseline and the ground truth is 0.35 pp. In contrast, the proposed model shows a much smaller average absolute difference among vowel classes, which is only 0.23 pp. While for non-vowel classes, both models produce an average absolute difference of approximately 0.12 pp between the proportions of recognized phoneme classes and the ground truths.

#### G.1.2    VISUALIZATION OF VOWEL CLUSTERS

In order to gain a deeper understanding of the models' performance in recognizing vowels, we further conduct a cluster analysis. Specifically, we collect data from the last layer of both models by locating frames within the temporal span of any vowels when processing each utterance from the test set. Each latent vector inferred by the last layer of a model, corresponding to these vowel frames, is treated as a data point for clustering. Note that one vowel can span multiple frames. Fig. 11 (a) and Fig. 11 (b) present the t-SNE results for the vowel latent vectors obtained from the wav2vec2 baseline and the proposed w20-t2f4 model, respectively. In the t-SNE result for the baseline, there is a non-negligible proportion of scattered points (as circled out in Fig. 11 (a)), which are distant from any clusters and are interleaved with each other in a sparse region of the 2-dimensional embedding space. This suggests that the baseline may struggle to correctly classify the vowel frames corresponding to these data points, which is likely because the representations abstracted by the last layer of the baseline lack sufficient information to distinguish between all vowel classes. On the contrary, in the t-SNE result for the proposed model which benefits from the incorporation of additional relational information from the local context, interleaving among data points from different vowel classes in the embedding space is greatly reduced, compared to that for the baseline. At the same time, data points from the same vowel class are still tightly clustered together, indicating better separability of vowel classes in the proposed model's representations.

Table 4: Proportions of phoneme classes recognized by baseline and proposed models (%). Values within parentheses indicate the absolute differences (pp) between the proportions recognized by the two models and the ground truth proportions.

| phoneme | | recognized by baseline | recognized by proposed | ground truth |
|---|---|---|---|---|
| vowel | aa | 3.50 (0.14) | 3.01 (0.63) | 3.64 |
| | ae | 2.47 (0.23) | 2.51 (0.28) | 2.24 |
| | ah | 5.18 (1.46) | 3.59 (0.14) | 3.72 |
| | aw | 0.15 (0.19) | 0.37 (0.02) | 0.34 |
| | ay | 1.40 (0.04) | 1.26 (0.10) | 1.35 |
| | eh | 1.94 (0.35) | 1.79 (0.50) | 2.29 |
| | er | 2.63 (0.84) | 3.27 (0.20) | 3.47 |
| | ey | 0.82 (0.46) | 1.22 (0.07) | 1.28 |
| | ih | 6.82 (0.58) | 7.03 (0.37) | 7.40 |
| | iy | 4.03 (0.27) | 4.79 (0.48) | 4.31 |
| | ow | 1.26 (0.02) | 1.33 (0.10) | 1.24 |
| | oy | 0.40 (0.02) | 0.48 (0.06) | 0.42 |
| | uh | 0.19 (0.17) | 0.14 (0.21) | 0.35 |
| | uw | 1.07 (0.12) | 1.29 (0.09) | 1.19 |
| | average | (0.35) | (**0.23**) | |
| non-vowel | b | 1.42 (0.01) | 1.57 (0.16) | 1.41 |
| | ch | 0.39 (0.02) | 0.44 (0.03) | 0.41 |
| | d | 1.83 (0.15) | 2.07 (0.09) | 1.98 |
| | dh | 1.66 (0.01) | 1.82 (0.14) | 1.67 |
| | dx | 1.96 (0.46) | 1.55 (0.05) | 1.49 |
| | f | 1.45 (0.00) | 1.46 (0.01) | 1.45 |
| | g | 1.23 (0.03) | 1.32 (0.12) | 1.20 |
| | hh | 1.03 (0.13) | 1.01 (0.15) | 1.15 |
| | jh | 0.64 (0.05) | 0.64 (0.05) | 0.59 |
| | k | 2.60 (0.04) | 2.77 (0.21) | 2.57 |
| | l | 4.26 (0.03) | 4.58 (0.29) | 4.29 |
| | m | 2.60 (0.10) | 2.59 (0.09) | 2.50 |
| | n | 5.02 (0.07) | 5.03 (0.08) | 4.95 |
| | ng | 0.79 (0.12) | 0.71 (0.05) | 0.67 |
| | p | 1.47 (0.05) | 1.52 (0.00) | 1.52 |
| | r | 4.21 (0.19) | 4.19 (0.17) | 4.01 |
| | s | 4.60 (0.40) | 4.45 (0.25) | 4.20 |
| | sh | 1.30 (0.08) | 1.35 (0.03) | 1.38 |
| | sil | 20.33 (0.02) | 19.70 (0.61) | 20.31 |
| | t | 2.86 (0.42) | 2.57 (0.13) | 2.44 |
| | th | 0.30 (0.12) | 0.37 (0.05) | 0.42 |
| | v | 1.05 (0.07) | 1.15 (0.02) | 1.13 |
| | w | 2.13 (0.16) | 1.99 (0.02) | 1.97 |
| | y | 1.11 (0.10) | 0.97 (0.03) | 1.01 |
| | z | 1.87 (0.15) | 2.14 (0.12) | 2.02 |
| | average | (0.12) | (0.12) | |

## G.2 ANALYSES OF LEARNED RELATIONAL INFORMATION IN PHONEME CLASSIFICATION TASKS

For a deeper understanding of how the learned relational information benefits the downstream phoneme recognition task, we further delve into a series of decomposed phoneme classification tasks and analyse the implications of the edges $\bar{\alpha}_{i,j}^{(t)}$ within the learned task-specific graphs.

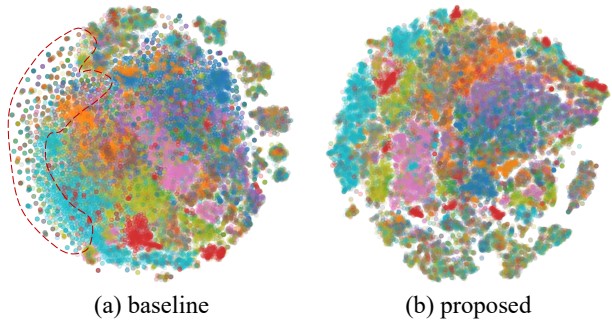

(a) baseline                          (b) proposed

Figure 11: t-SNE results for vowel latent vectors obtained from (a) baseline and (b) proposed model, respectively.

For each waveform segment that is aligned with a phoneme in the target sequence, we select 8 consecutive frames from its middle portion, and calculate the MFCCs $\mathbf{X}_{\mathrm{MFCC}} = [\mathbf{x}_{\mathrm{MFCC}}^{(i-7)}, \ldots, \mathbf{x}_{\mathrm{MFCC}}^{(i)}]$ as input for the phoneme classification task. The goal of this task is to predict the phoneme class of $\mathbf{X}_{\mathrm{MFCC}}$. This process is feasible since the TIMIT dataset provides annotations for the start and end instants of every phoneme within an utterance. We use the proposed spectro-temporal relational thinking module to calculate the graph embedding $\mathbf{r}$, followed by an MLP for predicting the phoneme class using $\tilde{\mathbf{x}}_{\mathrm{MFCC}}^{(i)} = [\mathbf{x}_{\mathrm{MFCC}}^{(i)T}, \mathbf{r}^T]^T$.

### G.2.1 VISUALIZATION OF LEARNED RELATIONAL INFORMATION

Regarding the phoneme classification task as a decomposed time step of the phoneme recognition task, we can derive a task-specific graph from the feature map $\mathbf{X}_{\mathrm{MFCC}}$ for each sample. For the ease of comparing the task-specific graphs derived from all samples, we flatten the edges $\bar{\alpha}_{i,j}$ into a $\binom{8}{2} = 28$ dimensional edge vector $\bar{\boldsymbol{\alpha}}$ for each graph. Fig. 12 visualizes the mean of edge vectors from each phoneme class by groups. The mixed group includes the approximants /w/ and /y/, as well as the liquids /l/ and /r/. As shown in each sub figure, the means of edge vectors from phoneme classes within the same group are relatively close to one another, indicating that phonemes from the same group, on average, share similar relational information. On the contrary, phonemes from different groups tend to contain distinct relational information, leading to larger distances between their corresponding means of edge vectors.

Next, we demonstrate the effectiveness of the learned relational information by conducting both a cluster analysis and a classification analysis using the edge vectors.

### G.2.2 CLUSTER ANALYSIS

We compute the t-SNE of the edge vectors $\bar{\boldsymbol{\alpha}}$ obtained from all the samples (Van der Maaten & Hinton, 2008), as shown in Fig. 13 (a). For phoneme groups with a sufficient number of samples (vowel, fricative, sil, as indicated in Table 5), the edge vectors are significantly clustered in the 2-dimensional embedding space. We also separately visualize the edge vectors of these three phoneme groups against the others in Fig. 13 (b)–(d), for the ease of observing group aggregations. For the edge vectors from the remaining phoneme groups, however, due to the limited number of samples, they do not show prominent aggregations. Nevertheless, this does not diminish the fact that the relational information involved in the learned graphs reveals similarities within phoneme groups and distinctions between phoneme groups.

### G.2.3 CLASSIFICATION ANALYSIS

We also use the edge vectors $\bar{\boldsymbol{\alpha}}$ obtained from all training samples to train a simple MLP classifier. This classifier predicts the phoneme group for an input edge vector, whose performance over the test samples is shown in Table 5. For phoneme groups with a sufficient number of samples, the classifier consistently achieves high precision. While due to significant group imbalance, it struggles to correctly identify the phoneme group of samples from the minority groups. Even though various

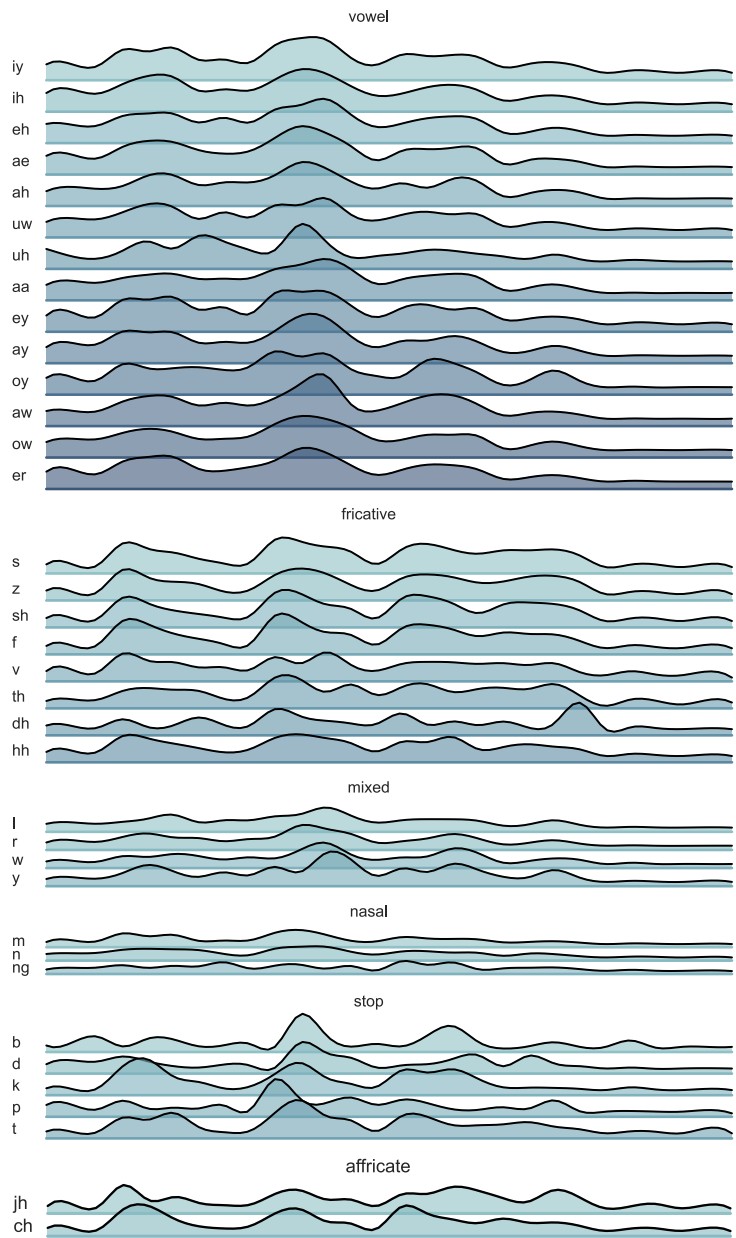

Figure 12: Visualization of learned relational information. The mean of edge vectors $\bar{\alpha}$ from each phoneme class is shown by groups. Resolution setting for time and frequency domains is (4, 2).

specialized approaches exist to mitigate the group imbalance problem (Johnson & Khoshgoftaar, 2019), addressing this issue is beyond the scope of our evaluation, which aims to assess the effectiveness of the learned relational information. Since the learned relational information (i.e., $\bar{\alpha}$) is the only input for the classifier, we reaffirm that this information is indeed effective and beneficial for the downstream task. The results obtained in Sections G.2.1–G.2.3 are highly consistent with one another.

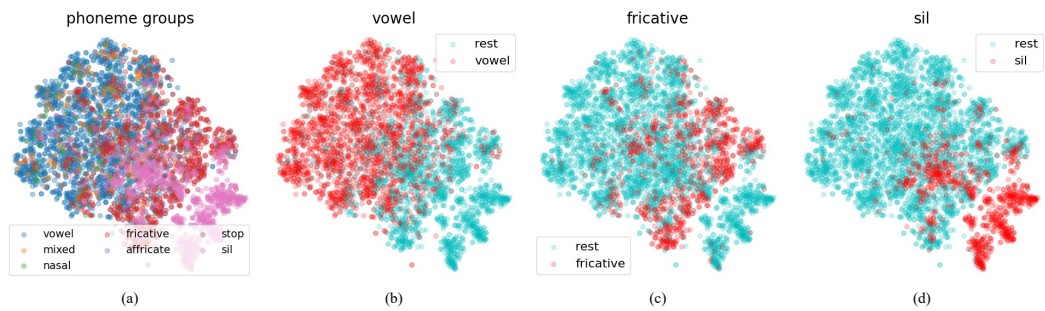

Figure 13: t-SNE results for edge vectors. (a) t-SNE result for edge vectors from all phoneme groups. (b)–(d) t-SNE results for edge vectors from vowel/fricative/sil vs. edge vectors from the rest phoneme groups, respectively.

Table 5: Performance of phoneme group classification with edge vectors in terms of precision (%).

| phoneme group | precision | number of samples |
|---|---|---|
| vowel | **84.69** | **15001** |
| fricative | **81.37** | **6538** |
| sil | **93.36** | **6748** |
| affricate | 0 | 252 |
| mixed | 0.88 | 1582 |
| nasal | 0 | 700 |
| stop | 0 | 308 |

Table 6: Speech recognition performances of baseline and proposed model in terms of WER (%) over TIMIT test set.

| | | w/o LM | 4-gram LM |
|---|---|---|---|
| baseline | wav2vec2 (Baevski et al., 2020) | 19.21 | 14.23 |
| proposed | 44-t4f2 | **18.72** | **13.77** |

### G.3 GENERALIZATION TO WORD-LEVEL RELATIONAL THINKING FOR SPEECH RECOGNITION

The proposed phoneme-level spectro-temporal relational thinking modeling can be readily generalized to modeling for other levels of linguistic units. Table 6 presents the speech recognition performance of a word-level relational thinking model built upon our proposed acoustic modeling framework over the TIMIT test set. In this model, we let $\mathbf{C}_t \in \mathbb{R}^{768 \times 44}$, spanning an average of 3 consecutive words. The kernel width and kernel stride for the temporal convolution in (1) are set to 9 and 5, respectively. The resolutions for time and frequency domains are set to (4, 2). We also include the performance of the state-of-the-art wav2vec2 baseline (Baevski et al., 2020) in the table for comparison. When language modeling is not incorporated, the proposed 44-t4f2 model outperforms the wav2vec2 baseline, achieving a 2.55% reduction in word error rate (WER). When a 4-gram language model is applied during decoding for both models, the proposed 44-t4f2 model exhibits an even greater performance gain, with a 3.23% decrease in WER.

