# OpenReview forum: "A Joint Spectro-Temporal Relational Thinking Based Acoustic Modeling Framework"
_ICLR.cc/2024/Conference — Submitted to ICLR 2024_

### Official Review · Reviewer_jL9N · 2023-10-29

**Soundness:** 3 good
**Presentation:** 3 good
**Contribution:** 3 good
**Rating:** 6
**Confidence:** 3

**Summary:**

This paper investigates injection of relational thinking on the frame-level phoneme recegnition task on TIMIT. The main proposal of the paper is that instead of focusing on time-only or frequency-only relations between consecutive frames, they should be jointly modeled. The acoustic model uses wav2vec2 features from audio and concatenates them with the features extracted from the relational thinking based graph embeddings before applying a classification layer. The model parameters are trained using a variational lower bound based approach. Experimental results show that joint time-frequency relations are important and the proposed method can outperform the wav2vec2 based baseline in phone recognition task on TIMIT. Analysis of the results show that the model is more effective on vowels as compared to the consonants.

**Strengths:**

- Originality:

1. The joint modeling of time-frequency seems to be effective on the phoneme recognition task. Additional analyses on the learned graphs show that the model can learn the vowel patterns more consistently.

2. It is nice to see the parallelism between human perception of vowels and the model’s results.

- Quality:
1. The paper has shown the model's effectiveness on the TIMIT task. The paper investigated various aspects of the model and design choices (even though they are sometimes limited).

- Clarity:
Clearly written

- Significance:
1. Even though the acoustic and graph embedding combination is performed in a rather straightforward way, the formulation of the learning objective can provide an opportunity for further extensions of the graph parameters.

**Weaknesses:**

1. More parameter settings and comparisons could have been investigated to strengthen the conclusions from the results.
2. Some additional analysis of the results could have been useful.

Please refer to the Questions below for details.

**Questions:**

1. Are the baselines trained with cross-entropy objective?

2. Would it make sense to Impose left to right constraint between the time steps for causality?

3. PER Analysis at the speaker level may give further intuition on how the model performs as compared to human perceptions.

4. It would be good to see Fig.5 repeated with t1f8 and t8f1 models.

5. Does feature mapping (Eq. 1) involve mixing of features within frequency bins? Is $  \Lambda $ diagonal or not?

6. Have you considered other types of spectra-temporal features for comparison? One example could be from, https://engineering.jhu.edu/lcap/data/uploads/pdfs/interspeech2012_carlin.pdf

7. Have you considered a comparison between spectro-temporal HMM based recognition and the proposed approach?

---

> ### Author Response · Authors · 2023-11-20
>
> We value the reviewer's insightful queries and the valuable feedback provided. Below are our responses addressing the questions and concerns raised by the reviewer.
>
> > 1. Are the baselines trained with cross-entropy objective?
>
> **AR:** No, the baselines used in our experiments were not trained with the cross-entropy objective. Instead, when we fine-tuned the baselines, we followed Baevski et al. (2020) and optimized the standard connectionist temporal classification (CTC) loss function (Graves et al., 2006). CTC is a commonly employed method for end-to-end sequence modeling tasks in the field of speech processing.
>
>
> > 2. Would it make sense to Impose left to right constraint between the time steps for causality?
>
> **AR:** We agree that it is essential to consider causality when modeling sequences. In our proposed relational thinking modeling, we have already taken causality into account. As stated at the beginning of Section 3.1, for each time step $t$, we constructed the feature map $\\mathbf{C}\_t$ by incorporating the acoustic feature vectors from the current time step and the previous $w - 1$ time steps as
> $$
> \\mathbf{C}\_t = [\\mathbf{c}\_{t - w + 1}, \\ldots, \\mathbf{c}\_t]. \\quad \\quad \\quad \text{(e)}
> $$
> This formulation (e) explicitly excluded acoustic feature vectors from future time steps. Essentially, this aligned with the concept of a left-to-right constraint by considering only past and current information and not incorporating information from future time steps.
>
>
> > 3. PER Analysis at the speaker level may give further intuition on how the model performs as compared to human perceptions.
>
> **AR:** We appreciate the reviewer's suggestion regarding PER analysis at the speaker level. It is indeed a valuable perspective for understanding how the model's performance compares to human perceptions. In this particular study, we adhered to the established protocol of reporting PER over the entire split to maintain consistency with previous research in phoneme recognition, ensuring a fair comparison. However, we acknowledge the significance of speaker-level analysis and plan to explore this approach in our future work.
>
>
> > 4. It would be good to see Fig.5 repeated with t1f8 and t8f1 models.
>
> **AR:** As indicated in Table 1, the two joint spectro-temporal models, w20-t4f2 and w20-t2f4, demonstrated superior performance compared to the purely temporal and spectral models, w20-t8f1 and w20-t1f8. This suggested that the spectro-temporal models had a better grasp of the inherent relational information involved in speech. Therefore, we believed it would be more beneficial to conduct further analyses on these joint spectro-temporal models rather than the purely temporal or spectral model to gain more accurate and valuable insights.
>
>
> > 5. Does feature mapping (Eq. 1) involve mixing of features within frequency bins? Is $\\mathbf{\\Lambda}$ diagonal or not?
>
> **AR:** As mentioned in Section 3.1, in our proposed framework, we utilized a temporal convolution to implement the filtering operator $\\Xi$ in (1). Therefore, it is possible that this operation may involve linear combinations of features within frequency bins. However, we can also choose to implement $\\Xi$ with various other methods (e.g., pooling), which may not mix features within frequency bins.
>
> $\\mathbf{\\Lambda}\_{t, d^{(f)}, d^{(t)}} \\in \\mathbb{R}^{D\_s \\times \\check{w}\_s}, \forall d^{(f)}, d^{(t)}$ are sub feature maps of $\\check{\\mathbf{C}}\_t$, which consist of the spectro and temporal domain feature information, and they are not necessarily diagonal.
>
>
> > 6. Have you considered other types of spectra-temporal features for comparison? One example could be from, https://engineering.jhu.edu/lcap/data/uploads/pdfs/interspeech2012\_carlin.pdf
>
> **AR:**  We thank the reviewer for directing us to the reference. Given that our paper primarily aimed to explore relational thinking modeling in recognition tasks, we anticipate that such modeling should be agnostic to acoustic feature types. Consequently, we utilized the state-of-the-art acoustic features extracted by wav2vec2. However, we acknowledge and appreciate the significance of considering various types of acoustic features, which can greatly influence system performance. To this end, we have performed additional experiments using MFCCs as an alternate acoustic feature type (as detailed in **A5: Generalization to Other Acoustic Features** in Section 5.1). As highlighted by the reviewer's reference, it is clear that their proposed spectro-temporal features exhibited similar performance to MFCCs in clean speech scenarios, which aligned with our study. These experiments with MFCCs yielded consistent results and further validated the generalizability of our proposed framework.

---

> > ### Author Response · Authors · 2023-11-20
> >
> > > 7. Have you considered a comparison between spectro-temporal HMM based recognition and the proposed approach?
> >
> > **AR:** We agree with the reviewer that noting the structural similarities, such a comparison would be interesting. However, given the substantial advancements in deep learning based models, especially those incorporating pre-trained encoders like wav2vec2, which have consistently outperformed HMM based models in phoneme recognition and speech recognition tasks, we did not prioritize a comparison between our proposed approach and HMM-based recognition. Furthermore, the primary focus of this study was to explore the effectiveness of relational thinking modeling, and we aimed to build upon the success of deep learning models in this context.

---

> > > ### Comment · Reviewer_jL9N · 2023-11-21
> > > **Read the Response, No Update to the Score**
> > >
> > > I would like to thank the authors for their response. I would like to keep my score as is.

---

### Official Review · Reviewer_KrqG · 2023-10-30

**Soundness:** 2 fair
**Presentation:** 2 fair
**Contribution:** 2 fair
**Rating:** 3
**Confidence:** 4

**Summary:**

This work proposed to use relational thinking-based acoustic model to learn the spectro-temporal correlation for automtic speech recognition task. Specficially, the proposed method is applied on the speech features extracted by a pre-trained wav2vec module. In the experiment, two tasks are performed including phenome recognition and automatic speech recognition. The results show the performance gain compared to these baseline systems.  These baseline systems are mostly the pre-training based methods which output the speech features.

**Strengths:**

The paper attempts to solve a valuable problem for acoustic modeling. The motivation of the work is clear and reasonable.

**Weaknesses:**

1, The innovation is not clear. The paper claims their innovation of using relational thinking based modeling method on spectro-temporal domain for acoustic modeling. However, from the description in the paper, there is no distinction between relational thinking based modeling and self-attention based modeling. For example, with several self-attention modeling layers stacked, it's equivalent to the so called relational thinking based model that pair-wised relation will be learnt among the transformed forms of each node (each time step), rather than the single step embedding. Therefore, theoretically, there is no difference between self-attention and relational thinking based method.
2, The experiment part is not complete. In order to demonstrate the superiority of the proposed relational thinking based method compared to the self-attention based method, the results of the self-attention should be included as one of the baseline results. However, the results of the paper only includes these feature extraction based method without extra modeling. In addition, the paper should also list the model size of each compared method to have a more fair comparision.
3, The tradeoff study between temporal context and spectral context is not able to lead such conclusion that higher frequency domain resolution provideds more benefits compared higher time domain resolution, as the results of these two setting are very close in the test set (20.80 vs. 20.66).

**Questions:**

1, Have you  done such experiment that replace the relational thinking based model with Transfomer/Conformer type of module? If so, what is the performance?
2, Could you please explain why the proposed method cannot perform well in non-vower recognition?

---

> ### Author Response · Authors · 2023-11-20
>
> We are grateful for the reviewer’s considerate questions and valuable feedback. Please find our responses to the questions and concerns raised by the reviewer below.
>
> > 1. The innovation is not clear. The paper claims their innovation of using relational thinking based modeling method on spectro-temporal domain for acoustic modeling. However, from the description in the paper, there is no distinction between relational thinking based modeling and self-attention based modeling. For example, with several self-attention modeling layers stacked, it's equivalent to the so called relational thinking based model that pair-wised relation will be learnt among the transformed forms of each node (each time step), rather than the single step embedding. Therefore, theoretically, there is no difference between self-attention and relational thinking based method.
>
> **AR:** Relational thinking and self-attention are two distinct mechanisms designed to capture different types of information. To illustrate the distinction, let us consider a simplified 2\-layer self-attention network w.l.o.g., where each layer $l$ comprises two nodes $\\mathbf{h}\_1^{(l)}$ and $\\mathbf{h}\_2^{(l)}$. According to (13), the calculation of nodes in the subsequent layer $l + 1$ is as follows:
> $$
> \\left [ \\mathbf{h}\_1^{(l + 1)}, \\mathbf{h}\_2^{(l + 1)} \\right ] = \\mathbf{W}\_v^{(l)} \\left [ \\mathbf{h}\_1^{(l)}, \\mathbf{h}\_2^{(l)} \\right ] \\left [
> \\begin{array}{cc}
> \\alpha\_{1, 1}^{(l)} & \\alpha\_{2, 1}^{(l)} \\\\
> \\alpha\_{1, 2}^{(l)} & \\alpha\_{2, 2}^{(l)}
> \\end{array}
> \\right ]. \\quad \\quad \\quad \\text{(b)}
> $$
> As a result, the state of a node $\\mathbf{h}\_2^{(3)}$ after undergoing two layers of self-attention calculations is
> $$
> \\begin{align}
> \\mathbf{h}\_2^{(3)} & = \\alpha\_{2, 1}^{(2)} \\mathbf{W}\_v^{(2)} \\left ( \\alpha\_{1, 1}^{(1)} \\mathbf{W}\_v^{(1)} \\mathbf{h}\_1^{(1)} + \\alpha\_{1, 2}^{(1)} \\mathbf{W}\_v^{(1)} \\mathbf{h}\_2^{(1)} \\right ) + \\alpha\_{2, 2}^{(2)} \\mathbf{W}\_v^{(2)} \\left ( \\alpha\_{2, 1}^{(1)} \\mathbf{W}\_v^{(1)} \\mathbf{h}\_1^{(1)} + \\alpha\_{2, 2}^{(1)} \\mathbf{W}\_v^{(1)} \\mathbf{h}\_2^{(1)} \right ) \\quad \\quad \\quad \\text{(c-1)} \\\\
> & = \\left ( \\alpha\_{1, 1}^{(1)} \\alpha\_{2, 1}^{(2)} + \\alpha\_{2, 1}^{(1)} \\alpha\_{2, 2}^{(2)} \\right ) \\mathbf{W}\_v^{(2)} \\mathbf{W}\_v^{(1)} \\mathbf{h}\_1^{(1)} + \\left ( \\alpha\_{1, 2}^{(1)} \\alpha\_{2, 1}^{(2)} + \\alpha\_{2, 2}^{(1)} \\alpha\_{2, 2}^{(2)} \\right ) \\mathbf{W}\_v^{(2)} \\mathbf{W}\_v^{(1)} \\mathbf{h}\_2^{(1)}. \\quad \\quad \\quad \\text{(c-2)}
> \end{align}
> $$
> Even though (c\-1) may exhibit a similar form to (12) (i.e., the graph embedding obtained by relational thinking, also as shown below),
> $$
> \\mathbf{r} = \\sum\_{(i, j) \\in \\{ (i, j) | i < j, (i, j) \\in \\bar{\\mathcal{E}} \\}} \\bar{\\alpha}\_{i, j} \\bar{f}\_{\\theta}(\\mathbf{h}\_{i}, \mathbf{h}\_{j}), \\quad \\quad \\quad \\text{(d)}
> $$
> particularly when we view $\\mathbf{W}\_v^{(2)} \\left ( \\alpha\_{1, 1}^{(1)} \\mathbf{W}\_v^{(1)} \\mathbf{h}\_1^{(1)} + \\alpha\_{1, 2}^{(1)} \\mathbf{W}\_v^{(1)} \\mathbf{h}\_2^{(1)} \\right )$ and $\\mathbf{W}\_v^{(2)} \\left ( \\alpha\_{2, 1}^{(1)} \\mathbf{W}\_v^{(1)} \\mathbf{h}\_1^{(1)} + \\alpha\_{2, 2}^{(1)} \\mathbf{W}\_v^{(1)} \\mathbf{h}\_2^{(1)} \\right )$ as node pair embedding functions from a linear family $\\mathcal{F} \\left ( \\mathbf{h}\_1^{(1)}, \\mathbf{h}\_2^{(1)} \\right )$, it is crucial to note that $\\mathbf{h}\_2^{(3)}$ is fundamentally still a weighted sum of node embeddings (as revealed by (c-2)) rather than a weighted sum of node pair embeddings as obtained by relational thinking (d), where $\\bar{f}\_{\\theta}(\\mathbf{h}\_{i}, \\mathbf{h}\_{j})$ is an arbitrary node pair embedding function. Therefore, the stacked self-attention mechanism and relational thinking are not entirely equivalent, and models cannot solely rely on the self-attention mechanism to effectively assess the importance of a pair of nodes, as discussed in Appendix D.
>
> Furthermore, it is also important to highlight that our proposed relational thinking modeling is a variational approach, whereas the self-attention mechanism is a deterministic approach. As elucidated in Nan et al. (2023), deterministic models can produce incorrect predictions when confronted with unseen data, particularly due to the discontinuous and sparse nature of the latent space. In contrast, variational models address this challenge by mapping the input data to collections of distributions in the latent space rather than singular points. By capturing uncertainty in the latent space, variational modeling enhances model generalization, rendering the model more adaptable to diverse and unseen data. Also, we have further clarified our novelty; please refer to our responses to Reviewer Q5NM for more details.

---

> > ### Author Response · Authors · 2023-11-20
> >
> > > 2. The experiment part is not complete. In order to demonstrate the superiority of the proposed relational thinking based method compared to the self-attention based method, the results of the self-attention should be included as one of the baseline results. However, the results of the paper only includes these feature extraction based method without extra modeling. In addition, the paper should also list the model size of each compared method to have a more fair comparision.
> >
> > **AR:** Our experiment **A4: Comparison with SOTA** in Section 5.1 indeed compared our proposed model with the self\-attention mechanism based baseline. The majority of state\-of\-the\-art systems in speech processing have been developed using transformer architectures (Vaswani et al., 2017), which fundamentally rely on self-attention mechanisms. Our baseline, wav2vec2 (Baevski et al., 2020), is just one example of these transformer\-based systems. By comparing these two models, we were in fact allowed to evaluate the additional benefits provided by relational thinking modeling in enhancing speech representation.
> >
> > Additionally, in our revised version of the paper, we have listed the model size of each compared method, as can be found in Section 5.1. We also provide the total numbers of parameters for the models used in our experiments below:
> >
> > a) wav2vec2 baseline: 94.4M;
> >
> > b) proposed relational thinking based models: 100.8M.
> >
> >
> > > 3. The tradeoff study between temporal context and spectral context is not able to lead such conclusion that higher frequency domain resolution provides more benefits compared higher time domain resolution, as the results of these two setting are very close in the test set (20.80 vs. 20.66).
> >
> > **AR:** Our concluding remark regarding the trade-off between temporal context and spectral context was based on a comparison not only between w20\-t4f2 and w20\-t2f4 but also between w20\-t8f1 and w20\-t1f8. It is true that the performance differences between w20\-t4f2 and w20\-t2f4 were not sufficiently significant. We appreciate the reviewer's feedback and acknowledge this limitation.
> >
> > However, our intention was not to make a definitive conclusion based on the current results. Instead, we aimed to highlight the potential benefits of various strategies for resolution setting in both domains. To draw more robust and convincing conclusions, we recognize the need for a more comprehensive investigation of the impact of resolution settings on system performance, which we plan to explore in our future work.
> >
> > In the revised version of the paper, we have weakened our concluding remark to improve the clarity and accuracy of the description and better convey the experimental results.

---

> > > ### Author Response · Authors · 2023-11-20
> > >
> > > > **Questions:**
> > > > 1. Have you done such experiment that replace the relational thinking based model with Transfomer/Conformer type of module? If so, what is the performance?
> > >
> > > **AR:** We did not conduct any experiments in which the relational thinking module was replaced with a transformer/conformer type module. In fact, such experiments are not feasible due to the prevalent use of transformer/conformer based architectures in state\-of\-the\-art acoustic feature extraction methods, including the wav2vec2 that we employed in our proposed framework. These architectures are deeply integrated into the foundation of modern acoustic feature extraction, making it impractical to substitute them in our experiments.
> > >
> > > Instead, to assess the additional benefits offered by relational thinking beyond what the attention mechanism has achieved, we compared the performance of two models: 1) the wav2vec2 baseline, which incorporated a transformer\-based feature extraction module; 2) the proposed model, which utilized a transformer-based feature extraction module in conjunction with a relational thinking module. Obviously, introducing a model that replaces the relational thinking module with a transformer\-based module would be redundant. Such a model would essentially replicate the structure of the wav2vec2 baseline, defeating the purpose of our investigation into the benefits of relational thinking.
> > >
> > >
> > > > 2. Could you please explain why the proposed method cannot perform well in non-vower recognition?
> > >
> > > **AR:** As we stated at the beginning of Section 5.3, one potential reason why the proposed relational thinking based model cannot show significant advantages over the baseline in recognizing non\-vowels is that non\-vowel phonemes tend to have shorter durations compared to vowels. This shorter duration for non-vowels does not allow the relational thinking module to capture as much relational information, which could benefit the downstream recognition task.  The subsequent analyses conducted in this subsection partially supported our hypothesis.

---

### Official Review · Reviewer_Q5NM · 2023-10-31

**Soundness:** 2 fair
**Presentation:** 2 fair
**Contribution:** 2 fair
**Rating:** 3
**Confidence:** 5

**Summary:**

The authors present a  spectrotemporal relational thinking-based framework for acoustic modeling. The proposed framework improves upon the original relational thinking-based frame by extending the probabilistic graph modeling from the temporal domain to the frequency-temporal domain. The paper reports a 7.82% improvement in phoneme recognition over the state-of-the-art for TIMIT phoneme recognition task.

**Strengths:**

- Biological Inspiration and Acoustic Modeling: The exploration of biologically-inspired algorithms, such as relational thinking, in acoustic modeling is noteworthy. Given humans' inherent ability to process audio signals across both frequency and temporal domains, the extension of the original relational thinking network to a temporal-frequency domain seems  reasonable.

- Promising Results on TIMIT: Experimental results on the TIMIT dataset, though small, show promise against various baselines. Additionally, the detailed analysis and visualization of the generated graph and its relationship with different phoneme categories provide valuable insights.

**Weaknesses:**

- Incremental Technical Contribution: The technical developments in this work appear to be an incremental advancement from Huang et al. (2020). The main modification is the extension of the input from one dimension to two dimensions, followed by a direct application of the relational thinking network proposed by Huang et al.

- Dataset Limitations: The experiments rely heavily on the TIMIT dataset, which is relatively small in size. To firmly establish the proposed method's efficacy and robustness, it is imperative to test it on larger, more diverse datasets and under complex conditions.

**Questions:**

See weakness.

---

> ### Author Response · Authors · 2023-11-20
>
> > Incremental Technical Contribution: The technical developments in this work appear to be an incremental advancement from Huang et al. (2020). The main modification is the extension of the input from one dimension to two dimensions, followed by a direct application of the relational thinking network proposed by Huang et al.
>
> **AR:** In addition to extending the input from one dimension to two dimensions, our work differs from that of Huang et al. (2020) in three perspectives:
>
> a) Our focus differs from Huang et al. (2020) as we examined a distinct application, one that readily extends to scenarios where the input and output sequences are of different lengths. This is a common scenario in sequence modeling tasks like speech recognition.  To accommodate these real-world applications, we developed a novel loss function that can handle such cases. This stands in contrast to the loss function proposed by Huang et al. (2020), which is applicable only to cases where input and output sequences have equal lengths. This paper addressed the challenges that arise when dealing with input and output sequences of different lengths, a common scenario in sequence modeling tasks like speech recognition. To accommodate real-world applications, we developed a novel loss function that can handle such cases. This stands in contrast to the loss function proposed by Huang et al. (2020), which is applicable only to cases where input and output sequences have equal lengths.
>
> b) Our paper included a comprehensive series of analyses toward the recognition results and the learned relational information, providing valuable insights into human speech perception and comprehension. We uncovered the patterns involved in the captured relations (i.e., the edges $\bar{\alpha}^{(t)}_{i, j}$ of the task-specific graph), which exhibited more similarities for phoneme classes within the same sub-group, while showed significant variations between phoneme classes from different sub-groups. We also conveyed that relational thinking modeling primarily enhanced the system by improving its performance in recognizing vowels. These insights represent a novel and previously unexplored aspect of research.
>
> c) We conducted a theoretical analysis that explained the distinctions between relational thinking and the self-attention mechanism (in Appendix~D). Our analysis underscored that relational thinking captured a new type of (co-occurrence or pair-wise) information that could offer additional advantages beyond what the attention mechanism has achieved.
>
>
> > Dataset Limitations: The experiments rely heavily on the TIMIT dataset, which is relatively small in size. To firmly establish the proposed method's efficacy and robustness, it is imperative to test it on larger, more diverse datasets and under complex conditions.
>
> **AR:** As clarified in Section 4, we used the TIMIT dataset because this study primarily aimed to delve deeper into the concepts of relational thinking with speech understanding and recognition tasks at the fine-grained level, serving as an initial step toward a comprehensive understanding of how this can be applied to speech. Such an analysis is feasible exclusively with the TIMIT dataset, which stands out as one of the very few datasets that provide precise information about the start and end instants of each phoneme. This unique attribute allowed us to conduct more in-depth analyses of the recognition results. For instance, it allowed us to uncover the patterns involved in the captured relations (i.e., the edges $\bar{\alpha}^{(t)}_{i, j}$ of the task-specific graph). These patterns exhibited more similarities for phoneme classes within the same sub-group, while they showed significant variations between phoneme classes from different sub-groups (as detailed in Section 5.2 and Appendix G.2). It also facilitated our ability to convey that relational thinking modeling primarily enhanced the system by improving its performance in recognizing vowels (as detailed in Section 5.3 and Appendix G.1). Without the manually annotated labels in the TIMIT dataset, we would not have been able to attain these insights. With the refined knowledge and deep understanding, we would like to extend and corroborate our findings on larger datasets beyond phoneme recognition in the future.

---

> > ### Comment · Reviewer_Q5NM · 2023-11-23
> > **Thanks**
> >
> > Thanks for your efforts.  I agree with the authors that this work introduces a new perspective to applying DGP( et al. (2020)) to two-dimensional settings. However, the technical complexity of tackling such a setting seems straightforward and trivial. Extension to larger datasets is indeed essential and interesting to the processing community. Due to the above concerns, I decided to keep my original ratings.

---

### Official Review · Reviewer_4GN3 · 2023-11-01

**Soundness:** 2 fair
**Presentation:** 1 poor
**Contribution:** 2 fair
**Rating:** 3
**Confidence:** 3

**Summary:**

This paper describes an approach to representing smoothed spectrograms using a graph formulation where features are computed from pairwise interactions between spectrogram chunks. These features are then used for phoneme classification in TIMIT, where they show good performance, achieving 9.2% phoneme error rate on the TIMIT test set.

**Strengths:**

The experiments seem to show that the approach works well.

**Weaknesses:**

This paper was very difficult to read and understand. It uses many words with very suggestive connotations like "Thinking" in the title, "unconscious", "mental impressions", etc. without the necessary strong justification for invoking them in the setting of a machine learning paper. These words obscure what is actually going on in the approach and are not necessary.

The task of phoneme classification on TIMIT is very old and is a reasonable first step in demonstrating the promise of an approach, but is definitely not sufficient to show that a model is learning a reasonable representation. Additionally, while the proposed system's results are good on the task (9.2% PER on the test set from Table 2), the reported wav2vec 2.0 baseline numbers (9.98% PER on the test set) are not the numbers that are reported in that paper (8.3% PER on the test set). It is not clear where the 9.98% number comes from.

Figure 5 visualizes four relational graphs that show hard to interpret spectrogram pieces without axis labels conected by lines of varying weights. It is not clear which weights we should expect to be strong or weak, although some are strong and some are weak.

There is also an analysis of the proportion of frames in which each phoneme is predicted in figure 6, showing that the proposed system predicts phonemes with closer proportions to the ground truth than the baseline system of wav2vec 2.0, although it does not show error rates or accuracies for these predictions. It is not clear which phonemes are more accurately predicted, just which ones are more frequently predicted.

There are 13 pages of appendices and reading through all of it still does not explain all of the necessary details like explicitly stating the loss that is optimized.

**Questions:**

Where do the numbers in table 2 for wav2vec 2.0 come from?

What is the loss that is actually optimized and what parameters are adjusted to optimize that loss?

---

> ### Author Response · Authors · 2023-11-20
>
> We thank the reviewer for his insightful questions and feedback. Please see below for our responses to the reviewer's questions and concerns.
>
> > This paper was very difficult to read and understand. It uses many words with very suggestive connotations like "Thinking" in the title, "unconscious", "mental impressions", etc. without the necessary strong justification for invoking them in the setting of a machine learning paper. These words obscure what is actually going on in the approach and are not necessary.
>
> **AR:** The term "relational thinking" in neuroscience is commonly used to describe the human ability to handle abstract mental representations of the relationships among objects, attributes, and events (Alexander, 2016). This encompasses concepts such as "unconscious" and "mental impressions". While this is an established area of study in neuroscience, it is a relatively novel field of inquiry within the machine learning community, particularly concerning speech understanding and recognition tasks (Huang et al., 2020). We recognize the importance of striving for clarity and precision in scientific writing as highlighted by the reviewer. However, it may not be possible to clearly convey the essence of this concept without relying on the terminologies borrowed from neuroscience.
>
>
> > The task of phoneme classification on TIMIT is very old and is a reasonable first step in demonstrating the promise of an approach, but is definitely not sufficient to show that a model is learning a reasonable representation. Additionally, while the proposed system's results are good on the task (9.2% PER on the test set from Table 2), the reported wav2vec 2.0 baseline numbers (9.98% PER on the test set) are not the numbers that are reported in that paper (8.3% PER on the test set). It is not clear where the 9.98% number comes from.
>
> **AR:** In Baevski et al. (2020), the authors explicitly mentioned that two versions of the wav2vec2 models, wav2vec2 BASE and wav2vec2 LARGE, were implemented. The authors reported a PER of 8.3\% using wav2vec2 LARGE (in Table 3), while we adopted wav2vec2 BASE in our experiments (as highlighted in Appendix F.1.1 of our paper). The rationale behind this choice is that the TIMIT dataset is not particularly large, and the wav2vec2 BASE, which has the same architecture but fewer parameters, is considered sufficient for the task at hand. While Baevski et al. (2020) did not provide the performance of wav2vec2 BASE, we fine-tuned it for our experiments, with the corresponding PER being 9.98\%.
>
> As clarified in Section 4, we used the TIMIT dataset because this study primarily aimed to delve deeper into the concepts of relational thinking with speech understanding and recognition tasks at the fine-grained level, serving as an initial step toward a comprehensive understanding of how this can be applied to speech. Such an analysis is feasible exclusively with the TIMIT dataset, which stands out as one of the very few datasets that provide precise information about the start and end instants of each phoneme. This unique attribute allowed us to conduct more in-depth analyses of the recognition results. For instance, it allowed us to uncover the patterns involved in the captured relations (i.e., the edges $\bar{\alpha}^{(t)}_{i, j}$ of the task-specific graph). These patterns exhibited more similarities for phoneme classes within the same sub-group, while they showed significant variations between phoneme classes from different sub-groups (as detailed in Section 5.2 and Appendix G.2). It also facilitated our ability to convey that relational thinking modeling primarily enhanced the system by improving its performance in recognizing vowels (as detailed in Section 5.3 and Appendix G.1). Without the manually annotated labels in the TIMIT dataset, we would not have been able to attain these insights. With the refined knowledge and deep understanding, we would like to extend and corroborate our findings on larger datasets beyond phoneme recognition in the future.

---

> > ### Author Response · Authors · 2023-11-20
> >
> > > Figure 5 visualizes four relational graphs that show hard to interpret spectrogram pieces without axis labels conected by lines of varying weights. It is not clear which weights we should expect to be strong or weak, although some are strong and some are weak.
> >
> > **AR:** Fig. 5 was primarily intended to visualize the captured relational information through our proposed relational thinking modeling and to highlight distinctions between relational thinking and the self-attention mechanism (e.g., the importance of the co-occurrence of a pair of nodes vs. the importance of a singular node). However, it was not our intention to use this figure to explicitly convey which weights should be strong or weak.
> >
> > Instead, the specific information about which weights should attain high values and which should attain low values was presented by Fig. 12. This figure offered insights into the connections between the learned edge values $\\bar{\\alpha}^{(t)}\_{i, j}$ and specific phoneme sub\-groups. In particular, it illustrated that the patterns of the edges exhibited greater similarities for phoneme classes within the same sub\-group, while they exhibited significant variations for phoneme classes from different sub\-groups. Therefore, by referring to Fig. 12, it becomes possible to tell the distributions of the edge values $\\bar{\\alpha}^{(t)}\_{i, j}$ within a task-specific graph associated with a particular phoneme sub-group. More details regarding this analysis can be found in Appendix G.2.
> >
> > In our revised version of the paper, we have made improvements to Fig. 5 by including time and frequency axes. We have also refined the caption of the figure to enhance its clarity and provide a clearer context for the indications within the figure.
> >
> >
> > > There is also an analysis of the proportion of frames in which each phoneme is predicted in figure 6, showing that the proposed system predicts phonemes with closer proportions to the ground truth than the baseline system of wav2vec 2.0, although it does not show error rates or accuracies for these predictions. It is not clear which phonemes are more accurately predicted, just which ones are more frequently predicted.
> >
> > **AR:** It is impossible to accurately derive the error rate for each phoneme class from the recognition results. Instead, to provide an alternative reflection of the models' performance in recognizing each phoneme class, we presented the proportions of recognized phoneme classes in Fig. 6 (along with Table 4). However, our primary focus in Section 5.3 was rather to understand how the proposed model contributes to the enhancement of phoneme recognition performance, and the most significant conclusion we reached in this subsection was that we confirmed relational thinking modeling primarily improved the model's ability to recognize vowels.
> >
> > As for the challenge in calculating error rates at the individual phoneme class level, it arises from the complex nature of recognition errors. To be specific, the output sequence produced by the model can contain various types of errors, including substitution errors, insertion errors, and deletion errors. These errors make it difficult to precisely align the output sequence with the ground truth. For example, consider a scenario where the model's output sequence is [/ix/, /ix/], while the ground truth is [/ah/, /ix/, /ae/]. In such a case, it is difficult to determine whether /ah/ has been incorrectly recognized as the first /ix/, or if /ae/ has been mistakenly identified as the second /ix/ in the output sequence. Therefore, it is not feasible to calculate the error rate for each phoneme, also reflected in the absence of such data in the literature.
> >
> > Also, note that analyzing the proportions of recognized phoneme classes was just the initial phase of our comprehensive analysis in Section 5.3. The key observations from this phase served as a foundation for the overall analysis, which helped us better understand how we initially leveraged prior knowledge/intuitive impressions and ultimately identified the source of performance improvement offered by the proposed model. Specifically, the proportions of recognized phoneme classes by the baseline and the proposed model against the ground truth proportions provided significant implications on the distinctions in behavior between the two models. This initial observation strongly motivated us to come up with the subsequent phase of analysis, where we separately evaluated the performance of the two models in recognizing vowels and non-vowels, ultimately leading to our conclusive findings.

---

> > > ### Author Response · Authors · 2023-11-20
> > >
> > > > What is the loss that is actually optimized and what parameters are adjusted to optimize that loss?
> > >
> > > **AR:** We explicitly defined the loss function for our proposed framework and provided a detailed explanation of how we derived a tractable form of this loss function in Section 3.2 (with supplementary information available in Appendix E).
> > >
> > > Our initial objective was to optimize the model log-likelihood $\\log p(y | \\mathcal{C})$. To make it tractable, we first derived the variational lower bound (Sohn et al., 2015) of it as
> > > $$
> > > \\log p(y | \\mathcal{C}) \\geq \\mathbb{E}\_{ q ( \\tilde{\\mathcal{A}}, \\mathcal{S} | \\mathcal{C} ) } [\\log p ( y | \\mathcal{C}, \\tilde{\\mathcal{A}}, \\mathcal{S} ) ] - \\sum\_{t = 1}^T \\text{div} ( q ( \\tilde{\\mathbf{A}}_t, \\mathbf{S}_t | \\mathbf{C}\_t ) \\| p ( \\tilde{\\mathbf{A}}\_t, \\mathbf{S}\_t | \\mathbf{C}\_t ) ) = \\mathcal{L}. \quad \quad \quad \text{(a)}
> > > $$
> > > In (a), the first term $\\mathbb{E}\_{ q ( \\tilde{\\mathcal{A}}, \\mathcal{S} | \\mathcal{C} ) } [\\log p ( y | \\mathcal{C}, \\tilde{\\mathcal{A}}, \\mathcal{S} ) ]$ was optimized with the connectionist temporal classification (CTC) loss (Graves et al., 2006), which is generally applied in speech recognition tasks (as revealed by (17)). Every KL divergence $\\text{div} ( q ( \\tilde{\\mathbf{A}}_t, \\mathbf{S}_t | \\mathbf{C}\_t ) \\| p ( \\tilde{\\mathbf{A}}\_t, \\mathbf{S}\_t | \\mathbf{C}\_t ) )$ in the second term of (a) was further decomposed (following (18)) into a set of KL divergences between two Gaussian distributions (20) and a set of KL divergences between two Binomial distributions (21). Finally, by substituting (18)--(21) into (a), we obtained a closed-form loss function, by optimizing which the model log-likelihood $\\log p(y | \\mathcal{C})$ could be maximized.

---

> > > > ### Comment · Reviewer_4GN3 · 2023-12-04
> > > > **Response to rebuttal**
> > > >
> > > > I would like the thank the authors for their detailed rebuttal. I have read it and decided to keep my rating as it is. One point of clarification about computing per-phoneme accuracies is that it is standard in the ASR literature, where a minimum edit distance alignment is computed between the predicted sequence and the ground truth sequence and errors are tabulated based on this. While it is possible that multiple paths could have the same edit distance, in practice, this doesn't make a big difference to the statistics and is generally worth doing and insightful. I would encourage the authors to perform this analysis.

---

### Meta-Review · Area_Chair_Mv6w · 2023-11-30

**Metareview:**

This paper introduces an acoustic modeling framework based on spectro-temporal relational thinking.  Experiments on TIMIT show improved performance on phoneme recognition. Further analysis shows that the proposed technique improves the model's ability to recognize vowels.  Acoustic modeling inspired by human learning process is interesting and the work represents a good effort to that end.  Experimental results on TIMIT also appear to be supportive. The authors' rebuttal is meticulous and clears up most of the concerns raised by the reviewers.  However,  there are standing concerns.  For instance,  the proposed framework needs to be evaluated on more and larger datasets in order to verify its generalization.  The work reported here is an extension of (Huang et al. 2020) to include spectro-temporal correlation. It would be more convincing if the authors could report results on CHiME-2, CHiME-5 and RelationalSWB that are used in (Huang et al. 2020).  I suggest the authors make modifications accordingly and submit to another venue.

**Justification For Why Not Higher Score:**

Experiments need to improve. Only report results on TIMIT which is a small dataset. There are also some technical concerns that need to be addressed.

**Justification For Why Not Lower Score:**

N/A

---

### Decision · Program_Chairs · 2024-01-16

Reject